

# Calving front positions for Greenland outlet glaciers (2002–2021): a spatially extensive seasonal record and benchmark dataset for algorithm validation

**Xi Lu[1, 2], Liming Jiang[1, 2], Daan Li[3], Yi Liu[1, 2], Andrew J. Sole[4], Stephen J. Livingstone[4]**

[1] State Key Laboratory of Precision Geodesy, Innovation Academy for Precision Measurement Science and Technology, Chinese Academy of Sciences, Wuhan 430077, China.

[2] College of Earth and Planetary Science, University of Chinese Academy of Sciences, Beijing 100049, China.

[3] College of Urban and Environmental Sciences, Yancheng Teachers University, Yancheng 224002, China.

[4] School of Geography and Planning, University of Sheffield, Sheffield, S10 2TN, UK.

Corresponding author: Liming Jiang (jlm@whigg.ac.cn)

**Abstract.** Calving front positions of marine-terminating glaciers are a key indicator of variations in glacier dynamics, ice–ocean interactions, and serve as critical boundary conditions for ice sheet models. High-precision, long-term records of calving front variability are essential for understanding glacier recession and calving processes, improving mass loss estimates, and supporting the development and validation of robust automated front-tracking algorithms. However, existing datasets often exhibit limited spatial coverage, inconsistent temporal resolution, and heterogeneous delineation methods, which result in

variable accuracy and insufficient detail, reducing the performance and transferability of automated calving front detection. Here, we present a spatially extensive, high-accuracy dataset of glacier calving front positions across Greenland, intended as a benchmark for algorithm training, model–data integration, and studies of seasonal glacier dynamics. The dataset comprises approximately 12,000 manually delineated calving front positions for ~290 outlet glaciers from 2002 through 2021, extracted from multi-source satellite imagery (Landsat, Sentinel-1/2, MODIS, ENVISAT, and ERS). Delineations were conducted using

standardized workflows in the Google Earth Engine platform and ArcGIS, and each record is accompanied by comprehensive metadata, including acquisition date, digitization method, source imagery, and other relevant attributes. Positional accuracy was evaluated through comparison with high-resolution PlanetScope imagery and manually interpreted reference datasets, confirming high geometric fidelity with positional offsets ranging from about 40 to 100 m across representative glaciers,



depending on image resolution and terminus complexity. In contrast, automated products tend to show reduced accuracy in verification areas with complex terminus morphology, reflecting their high sensitivity to image quality, limited generalizability across heterogeneous geometries, and the absence of large-scale, high-precision training data. This dataset contributes to mitigating these challenges by providing dense, manually validated, high-precision observations across Greenland, serving as a robust benchmark for developing and validating automated front detection algorithms, refining boundary representations in ice sheet models, and advancing understanding of ice–ocean interactions. The dataset is publicly available at https://doi.org/10.5281/zenodo.16879054 (Xi et al., 2025).

## 1 Introduction

Mass loss from ice sheets remains a dominant driver of contemporary global sea level rise (Shepherd et al., 2018; Frederikse et al., 2020). For the Greenland Ice Sheet (GrIS), nearly half of this dynamic ice loss is attributed to frontal ablation at marine-terminating outlet glaciers, primarily through calving and submarine melting (Enderlin et al., 2014; Mouginot et al., 2019). The positions of calving fronts provide critical insights into glacier dynamics, ice–ocean interactions, and act as essential time-varying boundary conditions for ice sheet modeling (Moon and Joughin, 2008; Catania et al., 2018; Nick et al., 2013; Choi et al., 2021). Accurately delineated and temporally consistent records of calving front positions are therefore necessary for assessing frontal retreat, constraining frontal mass fluxes, and improving the predictive capabilities of both process-based and machine learning models (Andersen et al., 2019; Fürst et al., 2015).

Over the past two decades, several manually delineated datasets of glacier calving front positions derived from optical or radar satellite imagery have provided valuable insights into glacier retreat patterns and terminus variability across Greenland (Table 1) (Murray et al., 2015a; Wood et al., 2021; Andersen et al., 2019). However, most of these datasets were developed in the context of individual case studies or regional modelling efforts, and consequently cover only a subset of Greenland's outlet glaciers, with annual or sporadic sampling and varied delineation approaches (Cassotto et al., 2017; Kehrl et al., 2017; Carr et al., 2013; Fried et al., 2018; Howat and Eddy, 2017; Moon et al., 2015; Sakakibara and Sugiyama, 2019; Bevan et al., 2012). Such variability introduces inconsistencies in spatial coverage, temporal resolution, and interpretation standards, limiting their applicability for large-scale assessments and reducing their utility in training or validating automated detection algorithms. To improve spatial and temporal completeness, recent efforts have combined multiple sources into composite datasets (Greene et al., 2024; Goliber et al., 2022). While such efforts enhance coverage, they often inherit the heterogeneity and biases of their source material. For example, the TermPicks dataset (Goliber et al., 2022) integrates over 39,000 calving front traces contributed by multiple researchers, substantially enhancing data accessibility and enabling large-scale historical analyses. However, as noted by the authors, the study further identifies spatial biases in data coverage, with high trace density at well-studied glaciers and limited representation elsewhere. These limitations underscore the continued need for standardized, manually curated datasets to support both fundamental research on glacier dynamics and the advancement of emerging front delineation techniques.



In recent years, the increasing availability of high-resolution satellite imagery, combined with advances in computational capacity, has greatly accelerated the development of automated calving front delineation methods. In particular, machine learning and deep learning techniques have shown strong potential for extracting glacier termini from large-scale remote sensing archives, offering enhanced processing efficiency and scalability (Mohajerani et al., 2019; Baumhoer et al., 2019; Cheng et al., 2021; Zhang et al., 2023; Black and Joughin, 2023). These approaches are especially valuable for monitoring Greenland's marine-terminating glaciers, which exhibit highly nonlinear responses to climatic and oceanic forcing (Brough et al., 2023; Catania et al., 2020; Choi et al., 2021). Despite these advantages, the performance of automated approaches remains heavily dependent on the availability of high-quality, manually delineated training data (Cheng et al., 2021). Current algorithms struggle in complex scenes such as mélange-choked fjords, shadowed termini, low-contrast imagery, and heavily crevassed margins (Seale et al., 2011; Cheng et al., 2021). Even advanced methods, trained on more than 1,500 labelled fronts, have successfully classified only ~22,000 images from Greenland—representing a small subset of the over 400,000 images currently available (Cheng et al., 2021; Zhang et al., 2023). In addition, deep learning models trained on relatively small or regionally concentrated datasets often lack sufficient generalizability across the full range of glacier types and environmental conditions found in Greenland (Zhang et al., 2023). Thus, while automated approaches—particularly deep learning methods—provide scalable solutions for large-scale calving front monitoring, their accuracy and transferability remain limited by the scarcity of high-quality training and validation data. Until such models achieve greater robustness and generalizability, comprehensive manually delineated datasets remain indispensable for both scientific investigation and algorithm development (Moon et al., 2015; Baumhoer et al., 2019). However, most existing calving front products suffer from sparse temporal sampling, heterogeneous delineation protocols, or incomplete spatial coverage, reducing their suitability for supervised learning and consistent benchmarking. The continued lack of large-scale, uniformly processed datasets with broad spatial reach and seasonal resolution poses a significant barrier to the advancement and evaluation of reliable, generalizable detection algorithms. Further, the ongoing expansion of satellite-based observations over polar regions reinforces the urgent need for consistent and well-curated terminus datasets to complement standardized glaciological records such as surface velocity and elevation (Goliber et al., 2022).

There remains a critical need for standardized, manually curated datasets to guide algorithm training and benchmark performance across Greenland's diverse and often challenging glaciological environments. In response to these limitations, we introduce a new dataset comprising ~12,000 manually delineated calving front positions for ~290 outlet glaciers across GrIS from 2002 through 2021 (Fig. 1). By integrating multi-source satellite imagery and employing standardized delineation workflows, this dataset provides near-complete spatial coverage and captures both short-term fluctuations and long-term retreat trends. Beyond its observational utility, the dataset has also been applied as a time-varying boundary condition in high-resolution transient ice flow modeling, enhancing model-data integration (Lu et al., 2025). By filling critical gaps in spatial completeness, temporal frequency, and methodological consistency, this product offers a robust foundation for studies of glacier dynamics and mass balance, as well as for the development and validation of automated front-tracking algorithms.



**Table 1: Summary of publicly available glacier calving front datasets for the GrIS. Greene et al. (2024) provides mask-based products without glacier-specific delineations.**

| Production source | Glacier count | Time span | Temporal Resolution | Method |
|---|---|---|---|---|
| **This dataset** | ~290 | 2002-2021 | Seasonal to monthly | GEEDiT, ArcGIS |
| **Greene et al. (2024)** | — | 1985-2022 | Monthly | Interpolated Greenland-wide mask |
| **TermPicks** | 278 | 1916-2020 | Decadal to monthly | Data compilation |
| **Wood et al. (2021)** | 226 | 1992-2017 | Annual | Manual |
| **Fahrner et al. (2021)** | 224 | 1984-2017 | Annual | GEEDiT |
| **MEaSUREs** | 219 | 2015-2021 | Weekly to monthly | Manual |
| **Murray et al. (2015b)** | 199 | 2000-2010 | Annual | Manual |
| **CALFIN** | 65 | 1972-2019 | Sub-annual | Deep learning |
| **Andersen et al. (2019)** | 47 | 1999-2018 | Annual | Manual |





**Figure 1: Spatial overview of the outlet glaciers in Greenland included in this dataset. Central panel shows glacier locations grouped by drainage basin. Surrounding maps illustrate some examples of temporally resolved calving front positions (color-coded by year) derived from multi-source imagery.**



## 2 Data and methodology

### 2.1 Satellite imagery

To map seasonal and interannual variations in glacier termini across Greenland from 2002 through 2021, we used multi-source satellite imagery from both optical and synthetic aperture radar (SAR) sensors (Table 2). The primary datasets were Landsat-5, Landsat-7, Landsat-8 and Sentinel-2 optical imagery, and Sentinel-1 SAR data (Moon and Joughin, 2008; Howat et al., 2008; Bevan et al., 2012), all of which were accessed and directly visualized through the Glacier Extraction and Evaluation Dataset Tool (GEEDiT) (Lea, 2018) without additional preprocessing.

A total of 2,069 Landsat images were used, selected from the Google Earth Engine (GEE) platform (Gorelick et al., 2017) with cloud cover less than 30%. Pan-sharpened composites were displayed in GEEDiT to enhance spatial clarity and facilitate consistent manual delineation of glacier termini. Due to limited daylight conditions during polar winter, images acquired in spring were preferred over autumn to approximate winter glacier conditions. Landsat-7 images acquired after 2003 were affected by the Scan Line Corrector (SLC) failure (Storey et al., 2005), which introduced data gaps. In such cases, GEEDiT allowed the use of temporally adjacent auxiliary scenes (within ±15 days) to aid interpretation.

To increase seasonal completeness and spatial consistency after 2015, we incorporated 560 Sentinel-1 and Sentinel-2 scenes. Sentinel-1 Ground Range Detected (GRD) products (acquired in Interferometric Wide mode) were used to capture glacier fronts regardless of cloud cover or solar illumination (Baumhoer et al., 2019; Kehrl et al., 2017). Sentinel-2 Level-1C images were visually screened in GEEDiT (Lea, 2018), and only those with clear views of the terminus region were selected for digitization.

To improve temporal continuity for several representative glaciers, additional scenes from Moderate Resolution Imaging Spectroradiometer (MODIS) (MOD09GQ, 250 m) (Hall et al., 2002), ENVIronmental monitoring SATellite (ENVISAT) Advanced Synthetic Aperture Radar (ASAR) (Image Mode, ~8 m), and ERS-1/2 (European Remote Sensing satellite) SAR (Precision Image mode, ~12.5 m) (Rignot and Kanagaratnam, 2006) were manually downloaded and processed outside of GEEDiT. These data were particularly useful during extended cloud cover or in early years of the study period where optical imagery was limited. All supplementary imagery was georeferenced and digitized in ENVI and ArcGIS to ensure consistency with the main dataset. A summary of all sensors, resolutions, and acquisition timeframes is provided in Table 2.



**Table 2: Summary of satellite remote sensing imagery used for calving front position delineation.**

| Platform | Spatial resolution | Data level | Time range for employment | Providers |
|---|---|---|---|---|
| **Landsat-5** | 15 m (pan) | Level-1TP | 2006-2009 | USGS |
| **Landsat-7** | 15 m (pan) | Level-1TP | 2002 - 2014 | USGS |
| **Landsat-8** | 15m (pan) | Level-1TP | 2013 - 2014 | USGS |
| **MODIS** | 250 m (Band 1) | Level-2 | 2002 - 2014 | USGS |
| **Sentinel-1** | 10 m (IW mode) | Level-1 GRD | 2014 - 2021 | ESA |
| **Sentinel-2** | 10 m (Bands 2-4, 8) | Level-1C | 2015- 2021 | ESA |
| **ENVISAT** | ~8 m (Image Mode) | Level-1B | 2002-2011 | ESA |
| **ERS-1/2** | 12.5 m (Precision Image) | Level-1.5 | 2002-2011 | ESA |

## 2.2 Calving front delineation procedure

The delineation of glacier termini was performed through a standardized workflow integrating manual interpretation and semi-automated digitization, as illustrated in Fig. 2. The procedure comprises three primary components: (1) satellite image preparation, (2) terminus extraction and editing, and (3) quality control and metadata creation and integration.

For the majority of Landsat-5, Landsat-7, Landsat-8, and Sentinel-1/2 images, calving front delineation was conducted directly within GEEDiT, which enables efficient visual interpretation and shapefile export without additional preprocessing (Goliber et al., 2022; Lea, 2018). For sensors not accessible via GEEDiT—including MODIS, ENVISAT ASAR, and ERS SAR—data were manually downloaded and processed externally using the ENVI and ArcGIS platform. MODIS MOD09GQ daily imagery (Level-2 Gridded) was used as an auxiliary dataset during periods of limited optical coverage. The surface reflectance was atmospherically corrected and georeferenced. To enhance interpretability, we computed Normalized Difference Water Index (NDWI) from red and near-infrared bands. It is worth mentioning that MODIS, due to its low resolution, mainly serves as a reference for the calving front changes during the acquisition period of two high-resolution images, assisting in the identification of sudden calving. ENVISAT ASAR (Level-1B) and ERS-1/2 SAR (Level-1.5) scenes were downloaded from the ESA archives. For ENVISAT ASAR, radiometric calibration to sigma nought (dB), terrain correction using a DEM, and speckle filtering (Refined Lee) were applied to enhance interpretability. For ERS-1/2 precision image products, additional terrain correction and filtering were performed to ensure spatial consistency. All SAR scenes were reprojected to WGS84 (EPSG:4326) for integration with other datasets.

Calving front positions were identified manually by visually interpreting the boundary between grounded ice and open ocean or ice mélange. Digitization was conducted at the native resolution of each sensor. For optical imagery, pan-sharpened

scenes were used to enhance spatial detail. For radar imagery (e.g., Sentinel-1 and ASAR), water–ice contrast and fjord
geometry were used to guide delineation. In ambiguous cases—such as heavy mélange cover, shadowed areas, or low-contrast
scenes—adjacent scenes within a ±15-day window was consulted for cross-validation. All digitized calving fronts were
subsequently reviewed and, where necessary, manually adjusted to ensure spatial continuity and consistency across time steps.
A temporal plausibility check was conducted to flag unrealistic advances or abrupt reversals in calving front position. Position
failing this check was subjected to additional scrutiny and corrected where appropriate. Each calving front position was
assigned a full metadata record and all ice front data were compiled into a centralized, glacier-ID-based directory structure.

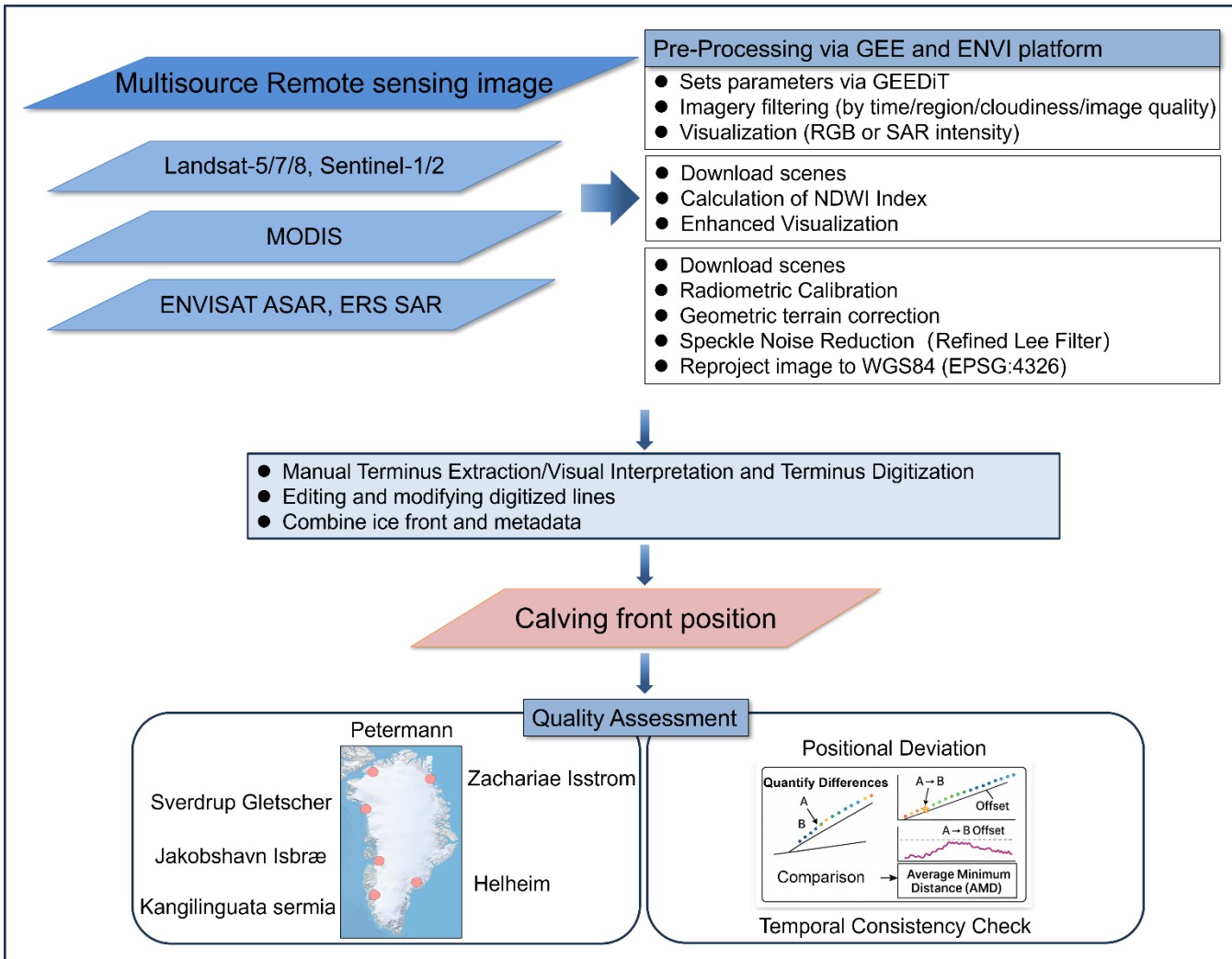

**Figure 2: Workflow of the calving front position delineation process. The system integrates multi-source satellite imagery through
direct access via GEE or manual preprocessing in ENVI and ArcGIS. Glacier fronts were manually delineated, followed by quality
assessment using offset metrics and cross-comparison with reference datasets.**



## 2.3 Validation

Digitization errors of glacier calving fronts are typically on the order of the source image resolution. For instance, calving front positions derived from Landsat 7 generally exhibit planimetric uncertainties of ~25 m (Moon et al., 2015). Beyond these resolution-based limitations, manual delineation can also be affected by scene-specific factors such as low contrast, shadows, mélange cover, and interpreter subjectivity. To assess the accuracy and consistency of our dataset, we conducted both qualitative and quantitative validations. These include (1) a visual assessment against high-resolution PlanetScope imagery to evaluate absolute geometric fidelity, and (2) a quantitative comparison with existing calving front datasets (e.g., TermPicks, AutoTerm) based on distance metrics to assess relative consistency across products (Fig. 2). Quality assessment was performed using distance-based metrics, including mean and median Average Minimum Distance (AMD), which quantifies the mean of the shortest Euclidean distances from each point on the reference line to the comparison line (Cheng et al., 2021) (Fig. 2), providing a robust measure of geometric similarity. As a purely geometric measure, AMD is calculated in absolute terms and does not reflect the direction of offset (i.e., it does not distinguish whether the comparison line lies up-glacier or down-glacier relative to the reference position). All validation efforts were based on availability-based spatio-temporal overlaps with external datasets and imagery, rather than targeted case selection, ensuring an objective and representative assessment.

PlanetScope imagery, acquired by a constellation of high-resolution Dove satellites operated by Planet Labs, provides near-daily observations at 3 m spatial resolution. Its fine spatial detail and temporal coverage make it a valuable benchmark for evaluating glacier front delineations. To assess the absolute positional accuracy of our dataset, we selected six representative glaciers across different regions of Greenland—Petermann, Zachariae Isstrøm, Helheim, Kangilinguata, Jakobshavn Isbræ, and Sverdrup glacier (Fig. 1). All selected Planet scenes were cloud-free and acquired on the exact same day of the corresponding image used for digitization. Rather than computing numerical offsets, this comparison focused on visual correspondence between our manually digitized fronts and the clearly identifiable glacier termini in the Planet imagery, with attention to morphological detail such as embayment and lateral margins.

In all six cases, the manually delineated fronts showed strong geometric agreement with Planet imagery, capturing key features with sub-pixel precision relative to the satellite imagery on which the delineations were originally based. Notably, the manual delineations remained reliable even in challenging visual conditions, such as shadowed regions (Fig. 3a) and heavily mélange-affected calving fronts (Fig. 3b–c, f), where frontal morphology remains ambiguous in moderate-resolution imagery. Where available, AutoTerm outputs were also included to show omissions or misalignments in automated results (e.g., missed segments or fragmented fronts), further underscoring the completeness and reliability of our manually derived dataset (Fig. 3a). Furthermore, the inclusion of Kangilinguata Glacier—a region not typically featured in other datasets—and additional comparisons not shown in Fig. 3 underscore the spatial coverage advantage of our dataset. These Planet-based comparisons confirm that our product achieves high absolute positional accuracy and geometric fidelity under diverse glaciological and imaging conditions, providing a robust foundation for the subsequent relative evaluation of large-scale calving front products.

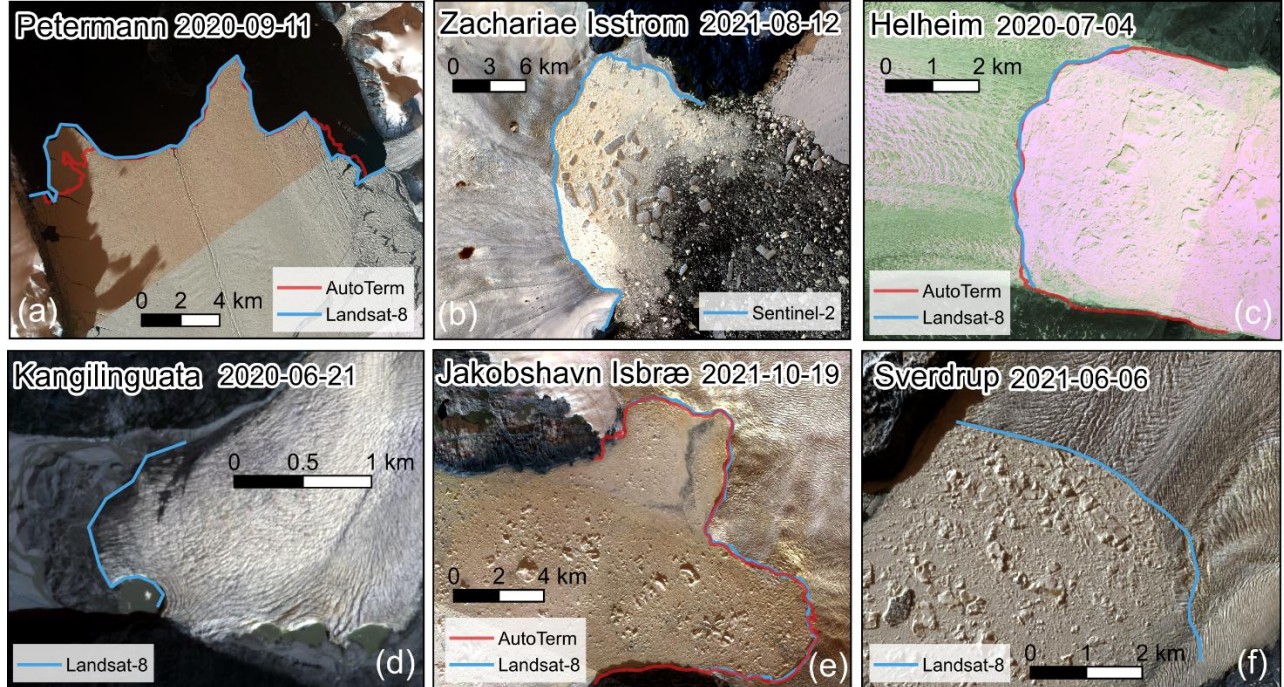

**Figure 3: Visual comparison of this manually delineated calving front (blue) with PlanetScope imagery for six representative glaciers across Greenland. Where available, AutoTerm results (red) are overlaid to highlight detection gaps or mismatches.**

To evaluate the consistency of this dataset relative to existing calving front products, we also compared the dataset presented in this study with several widely used, high-quality calving front products, including the time series compiled by TermPicks (Goliber et al., 2022), MEaSUREs (Black and Joughin, 2023), CALFIN (Cheng et al., 2021), and AutoTerm (Zhang et al., 2023). Due to the scale of our dataset, a glacier-by-glacier comparison with all prior products is infeasible. We therefore conducted comparisons for the representative outlet glaciers (Fig.1), selected to span multiple drainage basins and calving front types. Kangilinguata glacier, which is not included in any of the existing datasets, was excluded from this comparison. Where overlapping scenes were available, we compared manually delineated fronts in this study with those from each dataset using both visual overlays and distance-based metrics, with the AMD used to quantify spatial offset.

Figures 4-8 illustrate case studies for Petermann, Jakobshavn Isbræ, Zachariae Isstrøm, Helheim and Sverdrup glaciers, respectively. For each glacier, we present visual overlays of multi-source calving front positions and corresponding AMD-based offset profiles for three representative dates, along with a long-term time series comparison. Common findings across all sites show that this manually delineated calving fronts consistently outperform automated methods, particularly in visually complex regions. These include mélange-filled fjords, frontal rifting zones, shadowed margins, and lateral embayment, where automated algorithms often fail to preserve morphological detail or misplace the terminus. In earlier years or under challenging image conditions (e.g., mélange or cloud), AutoTerm and CALFIN exhibit large offset(>1km) and reduced morphological precision, often smoothing over frontal curvature or omitting finer-scale features. These deficiencies are especially pronounced in early Landsat scenes and low-contrast fjord settings. For example, at Petermann Glacier (Figs. 4a–c), irregular calving front





 At Jakobshavn Isbræ (Fig. 5a), lateral mélange accumulation

obscures the terminus, causing AutoTerm to deviate significantly from other datasets. A particularly notable error occurs in Fig. 5c, where shadowed side regions are misclassified as part of the calving front, resulting in the omission of a retreat signal during a known disintegration phase. For Zachariae Isstrøm (Figs. 6b–c), algorithmic misidentification of surface fractures as the calving margin inflates the spatial offset, particularly in zones where the actual terminus is fragmented or poorly defined. For Helheim (Figs. 7b–c), low image contrast and poorly defined lateral margins lead to deviations exceeding 2 km in the

automatically extracted fronts. In contrast, manual delineations capture subtle structures such as ice tongue protrusions and calving embayment concavities with greater fidelity, supporting their use as a reference for both model boundary conditions and algorithm training. These examples highlight common challenges in automated products when faced with ambiguous spectral signals or complex frontal configurations, and reinforce the added interpretive value of high-quality manual delineations under challenging observational conditions. The delineations produced in this study remain consistent and

structurally accurate even in these environments, highlighting the added value of expert interpretation.

In addition, our dataset exhibits strong consistency with other manually compiled calving front products in capturing both seasonal and interannual glacier dynamics. Time series comparisons with TermPicks (Goliber et al., 2022) and MEaSUREs (Black and Joughin, 2023) reveal close alignment across multiple glaciers and observation periods (Figs. 4-8g). For example, at Petermann Glacier, the long-term position change trends are highly consistent among the three datasets, confirming their

shared capacity to resolve broad-scale frontal evolution (Fig.4g). The time series in Fig.6g and Fig.8g further supports that close correspondence between our dataset and MEaSUREs across the observation period. Notably, the 2004–2005 calving event and the subsequent advance were also captured (Fig. 6g). Our product resolves seasonal fluctuations and short-term variability that are consistent with MEaSUREs during overlapping years, while also extending the record back to the early 2000s, providing enhanced temporal continuity for long-term monitoring of Zachariae Isstrom and Sverdrup glacier. Offset

profiles computed using the Average Minimum Distance (AMD) metric further substantiate this agreement. At Petermann, the AMD between our dataset and TermPicks and MEaSUREs is 202.2 m (2003), 49.3 m (2017), and 68.6 m (2020), respectively (Fig. 4d–f). The larger deviation in 2003 reflects the presence of mélange and complex frontal geometry, which challenge all methods. At Jakobshavn Isbræ (Fig. 5d–f), mean offsets range from 68.9 to 135.7 m, with the highest values again associated with earlier imagery and dynamic terminus changes. For Zachariae Isstrøm (Fig. 6d–f), AMD values range from 45.6 to 94.1 m

across three dates, with strong spatial alignment observed in clearer scenes such as 2018 (Fig. 6e). For Sverdrup Glacier (Figs. 8d–f), the AMD values ranged from 73.8 m to 110.7 m across the three observation dates, showing good overall agreement without significant deviations.

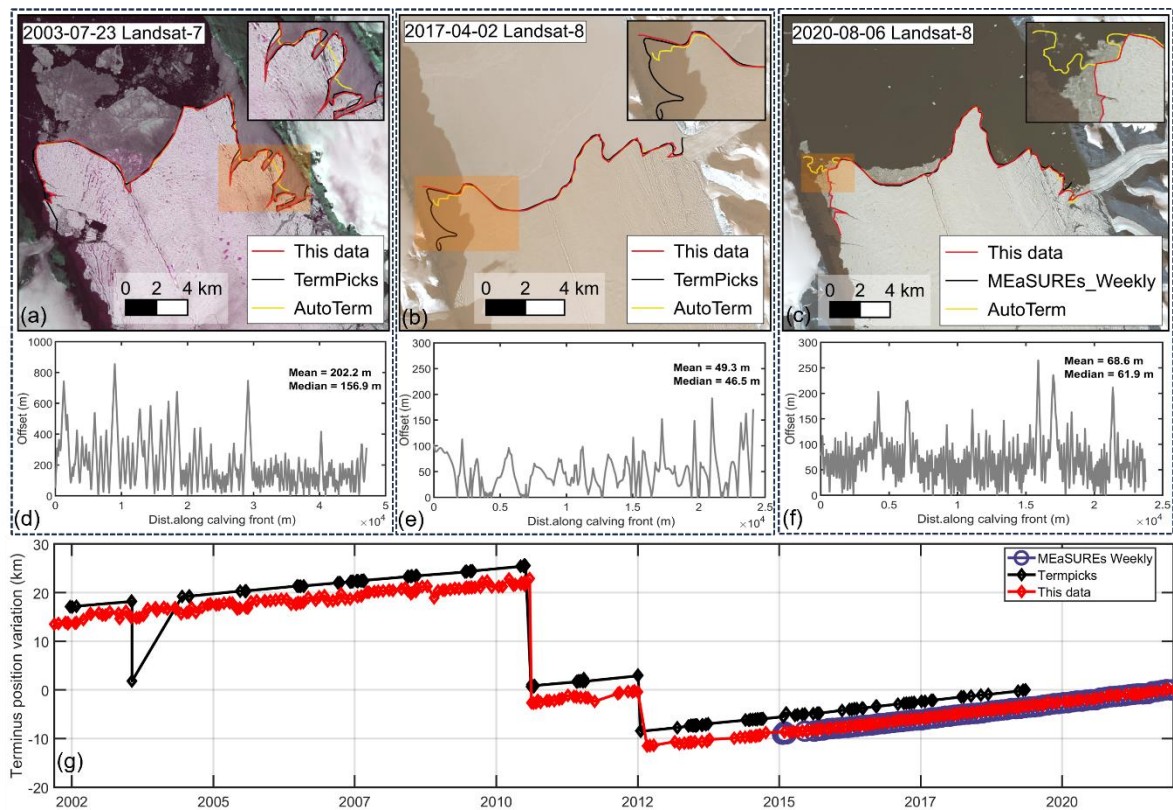

**Figure 4: Comparison of delineated calving fronts for Petermann Glacier on three dates (2003-07-23, 2017-04-02 and 2020-08-06).**
**(a–c) overlay this study's fronts (red) with TermPicks (black), AutoTerm (yellow) and MEaSUREs Weekly to Monthly (black) on**
**Landsat scenes; insets zoom in on areas of complex mélange cover or irregular geometry. (d–f) show the along-front offset profiles**
**(AMD) for each date. (d) and (e) show comparisons between this dataset and TermPicks, corresponding to (a) and (b), respectively.**
**(f) compares this dataset with MEaSUREs Weekly, corresponding to (c). Each profile displays both the average and median offsets.**
**(g) presents the time series of manually delineated calving front position variation (km) from 2002 through 2021.**

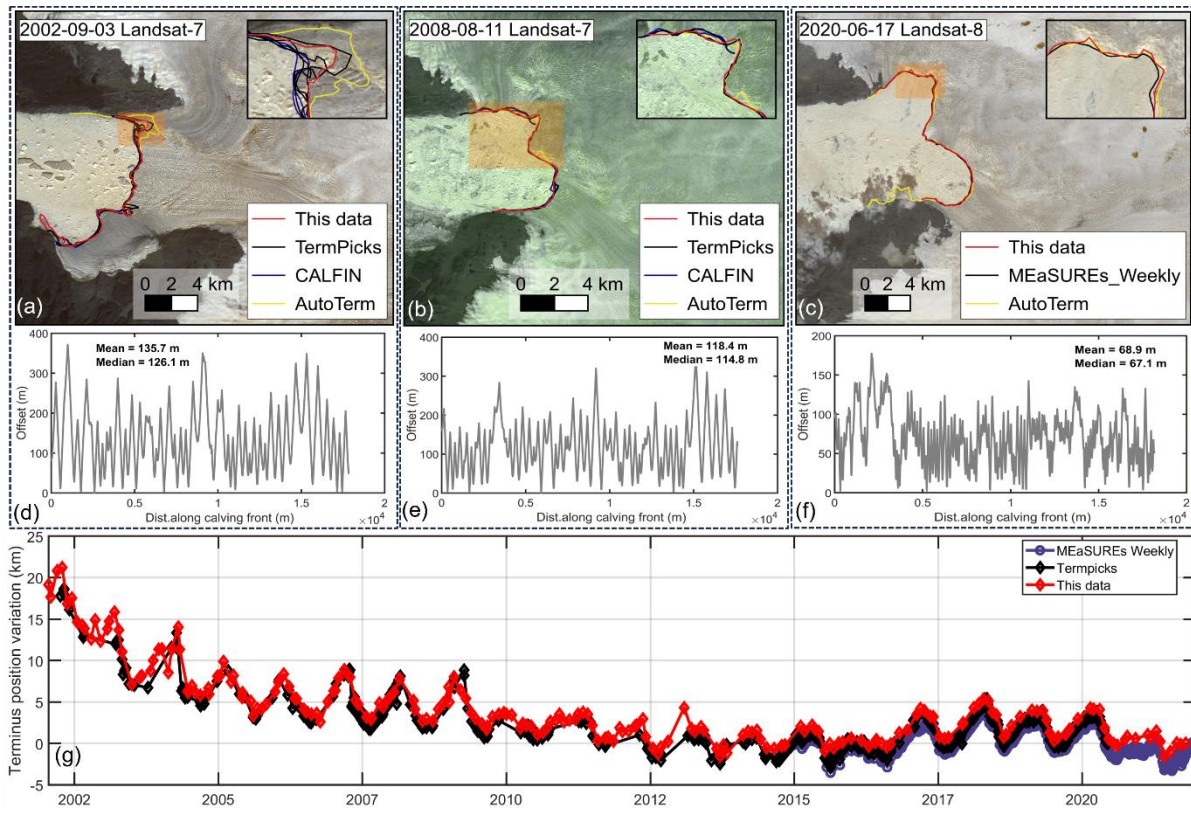

**Figure 5: Spatial and quantitative comparison of calving fronts for Jakobshavn Isbræ Glacier at three representative dates (2002-09-03, 2008-08-11, 2020-06-17). (a–c) show this study's delineations (red) alongside TermPicks (black), CALFIN (blue), AutoTerm (yellow) and MEaSUREs Weekly to Monthly (black). Insets highlight zones of mélange or lateral embayment complexity. (d–f) plot the along-front offset distributions. (d) and (e) show comparisons between this dataset and TermPicks, corresponding to (a) and (b), respectively. (f) compares this dataset with MEaSUREs Weekly, corresponding to (c). Each profile displays both the average and median offsets. (g) presents the time series of manually delineated calving front position variation (km) from 2002 through 2021.**

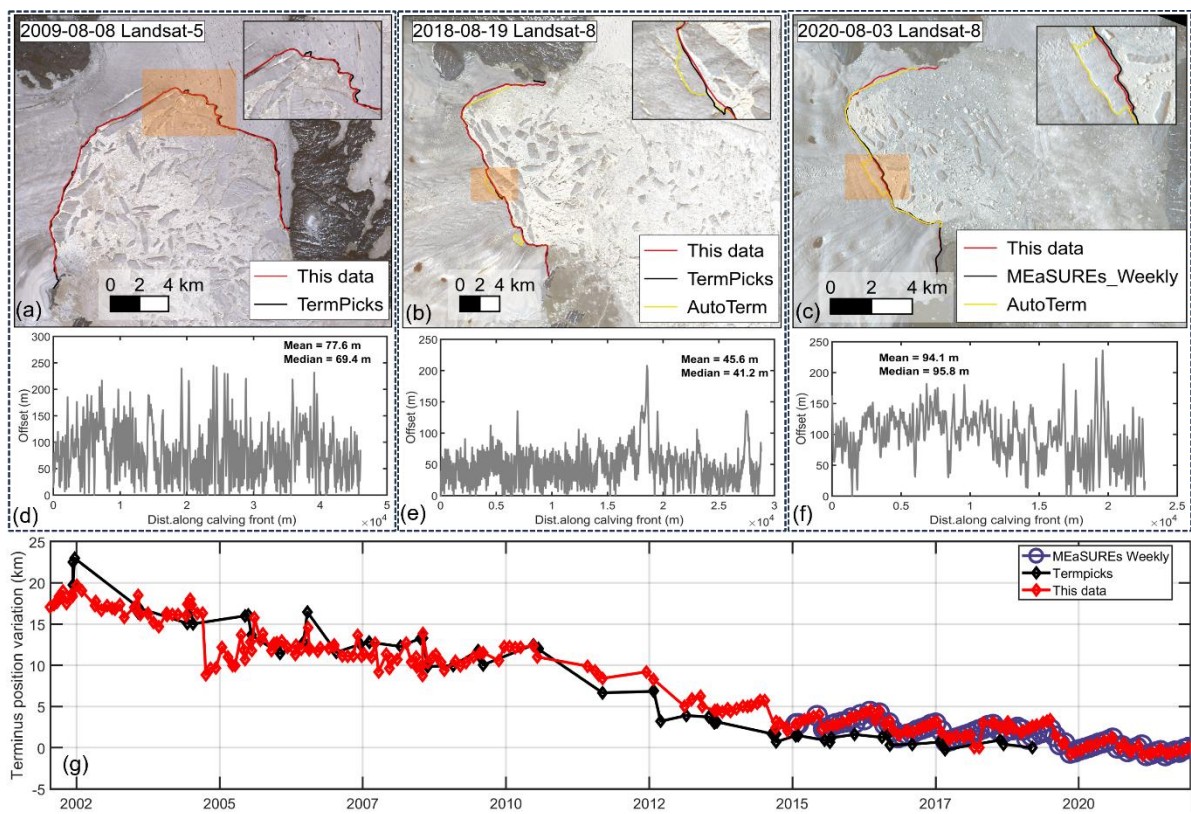

**Figure 6: Multi-temporal assessment of Zachariae Isstrom calving front positions on 2009-08-08, 2018-08-19 and 2020-08-03. (a–c)**
**overlay our manual fronts (red) with TermPicks (black), AutoTerm (yellow) and MEaSUREs Weekly to Monthly (black) on Landsat images; zoomed insets show areas of high morphological complexity. Panels (d–f) illustrate along-front offset profiles for each date. (d) and (e) show comparisons between this dataset and TermPicks, corresponding to (a) and (b), respectively. (f) compares this dataset with MEaSUREs Weekly, corresponding to (c). Each profile displays both the average and median offsets. (g) presents the time series of manually delineated calving front position variation (km) from 2002 through 2021.**


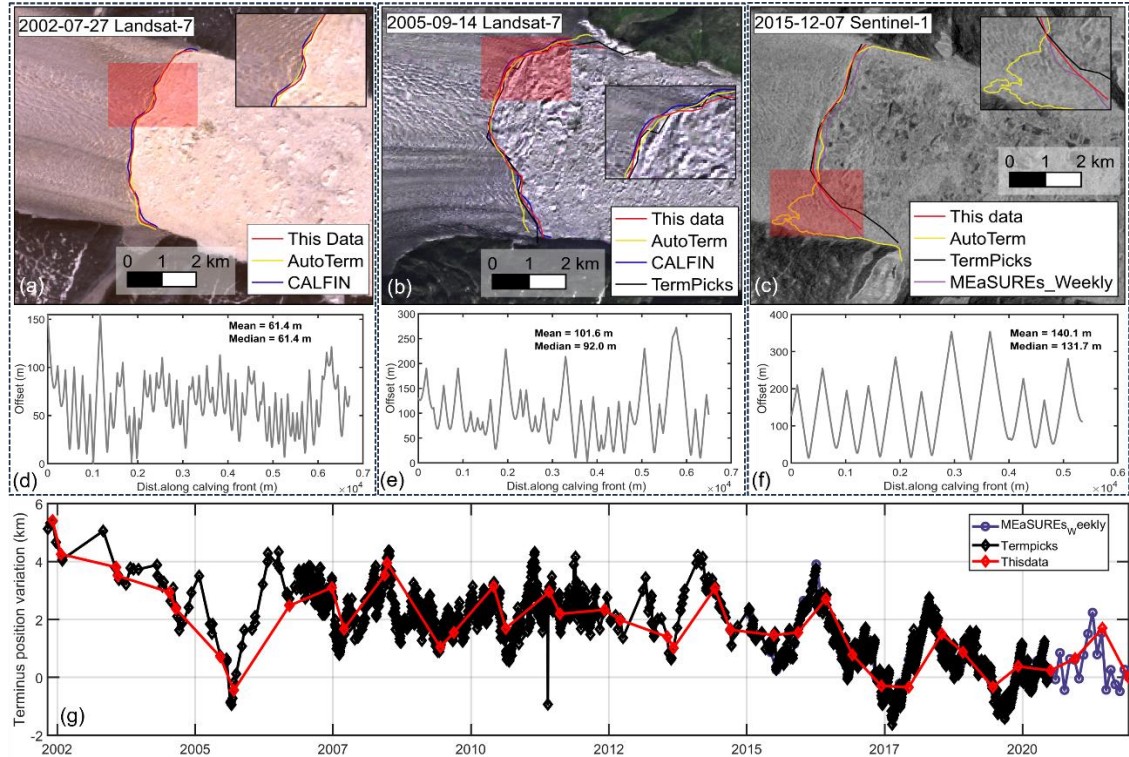

**Figure 7: Multi-temporal assessment of Helheim calving front positions on 2002-07-27, 2005-09-14 and 2015-12-07. (a–c) overlay our manual fronts (red) with TermPicks (black), CALFIN (blue), AutoTerm (yellow) and MEaSUREs Weekly to Monthly (purple) on Landsat and Sentinel-1 images; zoomed insets show areas of high morphological complexity. Panels (d–f) illustrate along-front offset profiles for each date. (d) show comparisons between this dataset and CALFIN, corresponding to (a). (e) and (f) compares this dataset with TermPicks, corresponding to (b) and (c), respectively. Each profile displays both the average and median offsets. (g) presents the time series of manually delineated calving front position variation (km) from 2002 through 2021.**

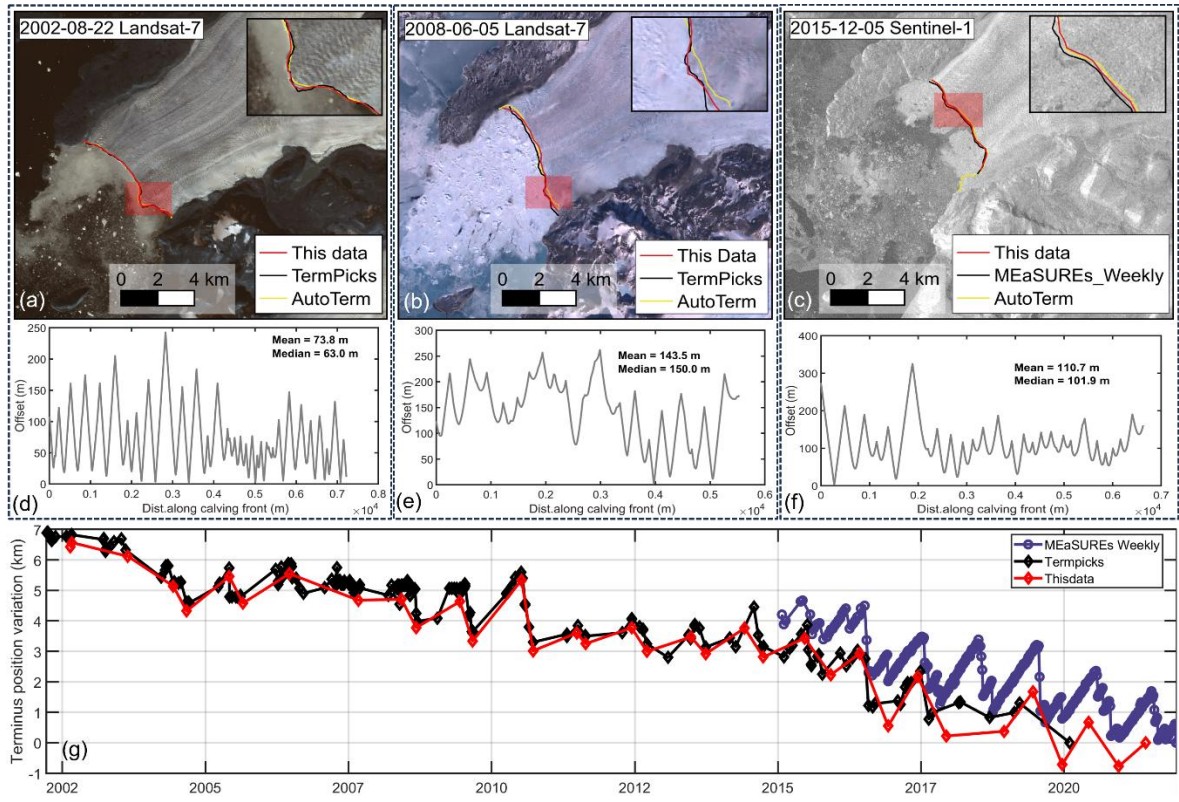

**Figure 8: Multi-temporal assessment of Sverdrup calving front positions on 2002-08-22, 2008-06-05 and 2015-12-05. (a–c) overlay our manual fronts (red) with TermPicks (black), AutoTerm (yellow) and MEaSUREs Weekly to Monthly (black) on Landsat and Sentinel-1 images; zoomed insets show areas of high morphological complexity. Panels (d–f) illustrate along-front offset profiles for each date. (d) and (e) show comparisons between this dataset and TermPicks, corresponding to (a) and (b), respectively. (f) compares this dataset with MEaSUREs Weekly, corresponding to (c). Each profile displays both the average and median offsets. (g) presents the time series of manually delineated calving front position variation (km) from 2002 through 2021.**

Across all comparative analyses, our manually delineated calving front dataset demonstrates high positional consistency with existing reference products, with mean offsets typically ranging from 40 to 100 m depending on glacier geometry, image quality, and surface contrast. While automated and semi-automated methods perform well in clear, high-contrast conditions, our product consistently outperforms them in more challenging environments—such as mélange-filled fjords, shadowed regions, and areas of low contrast—by preserving fine-scale morphological detail, including rifts, lateral embayments, and ice tongue structures that are often misrepresented or omitted by algorithmic approaches (Baumhoer et al., 2019).

In comparison with other manually curated datasets, such as TermPicks (Goliber et al., 2022) and MEaSUREs (Black and Joughin, 2023), this product captures similar seasonal and interannual trends but offers key advantages in spatial and temporal coverage. Specifically, it provides the longest continuous record to date (2002–2021), near-complete coverage of Greenland's marine-terminating outlet glaciers (~290), and a higher sampling frequency suitable for resolving short-term dynamic events. These strengths are further corroborated by high-precision agreement with PlanetScope imagery, where positional discrepancies are consistently below 10 m, despite the coarser resolution of our primary input data (10–30 m).



Overall, the validation results establish our dataset as a robust benchmark for glacier front monitoring and time series analysis. It offers not only a reliable foundation for modeling glacier dynamics at high resolution, but also a critical reference standard for training and validating automated calving front detection algorithms—particularly in observationally complex or poorly constrained regions.

## 3 Data product and usage notes

The dataset described in this study provides manually delineated calving front positions for ~290 outlet glaciers across Greenland, spanning the period 2002 to 2021. It includes approximately 12,000 individual calving front positions, offering broad spatial coverage and seasonal to sub-seasonal temporal resolution across nearly two decades (Fig. 9). The full dataset is publicly available at https://doi.org/10.5281/zenodo.16879054 (Xi et al., 2025).

The dataset is organized into a GeoPackage structure by glacier ID, with all delineations for a given glacier stored as an individual GeoPackage and named according to the AutoTerm (Zhang et al., 2023) convention to facilitate flexible queries, analysis, and integration with existing studies. For glaciers not included in AutoTerm, IDs were assigned based on the names of their nearest neighboring glaciers. For example, a glacier located near GID091 is named *New_NeighborGID091.shp*, indicating that it is a newly identified glacier front assigned based on its proximity to GID091. In addition, files named *New_NeighborGID091_x.shp* (where *x* = 1, 2, …) indicate multiple newly digitized calving fronts located near GID091. In total, 69 glaciers in this dataset have calving front positions that are newly recorded compared with those in the AutoTerm dataset. All spatial data are georeferenced in WGS 84 geographic coordinates (EPSG:4326), ensuring compatibility with common GIS and remote sensing tools. Each terminus is accompanied by structured metadata with the following attributes:

- ImagePath – source imagery used for digitization
- Satellite – platform name (e.g., Landsat-7, Sentinel-2)
- Method –digitization tool used (GEEDiT or ArcGIS)
- Date –acquisition date in the format YYYY-MM-DD
- Season – classified as 'Melt' (May–September) or 'Non-Melt' (all other months)

To support time series analysis, the metadata are formatted in a consistent structure, allowing users to easily filter or aggregate terminus records by date, season, satellite type, and other relevant attributes. Owing to variability in satellite coverage and cloud conditions, the temporal sampling density differs among glaciers. For most glaciers, typical intervals range from approximately three to five months (Fig. 9), whereas glaciers with higher scientific interest or active frontal variability—such as Jakobshavn Isbræ and Helheim—are captured at higher temporal frequency.



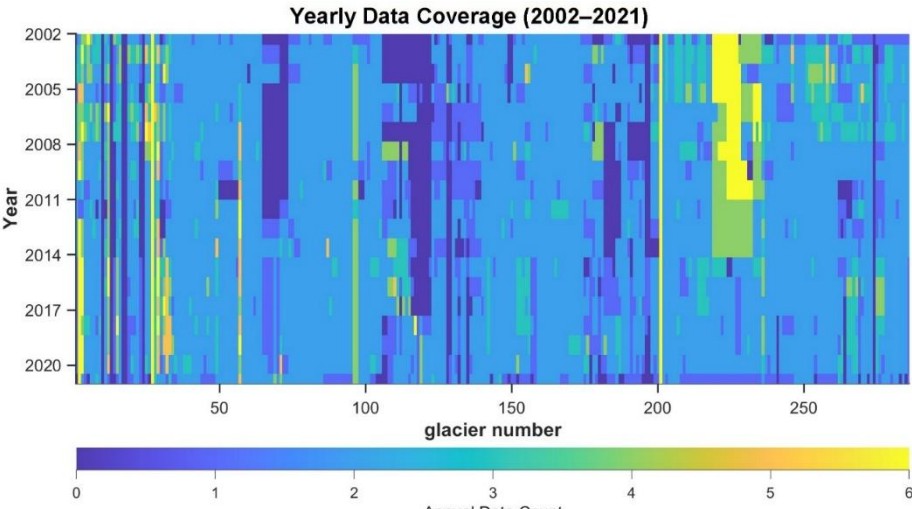

**Figure 9: Calving front positions delineation density (2002–2021) for ~290 Greenland outlet glaciers. Each column represents a glacier, each row a calendar year, and the colour scale indicates the number of manually digitized calving front positions per glacier per year.**

The dataset enables the analysis of both long-term retreat trends and seasonal calving front dynamics. Representative time series from individual glaciers highlight a diverse range of calving front behaviors, including episodic calving events (Fig. 4), progressive multi-year retreat (Fig. 6, 8), and regular seasonal advance–retreat cycles (Fig. 5, 7).

In addition to supporting high-resolution analyses of seasonal glacier dynamics, this dataset provides a comprehensive perspective on the spatial variability and long-term evolution of calving front positions across the GrIS, thereby offering valuable insights into large-scale patterns of glacier change. Fig. 10 presents the average calving front position change for glaciers in each of Greenland's drainage basins over the 2002–2021 period. Of the ~290 glaciers included, 84% experienced retreat from 2002 through 2021, with a mean retreat of 1.37 km across the ice sheet. Regional patterns reveal substantial spatial heterogeneity. The northwest (1.52 km), southeast (1.17 km), northeast (1.14 km), and west (1.13 km) basins exhibited the greatest mean retreat. In contrast, the northern (0.37 km) and southwestern (0.36 km) basins showed comparatively minor changes in calving front position.



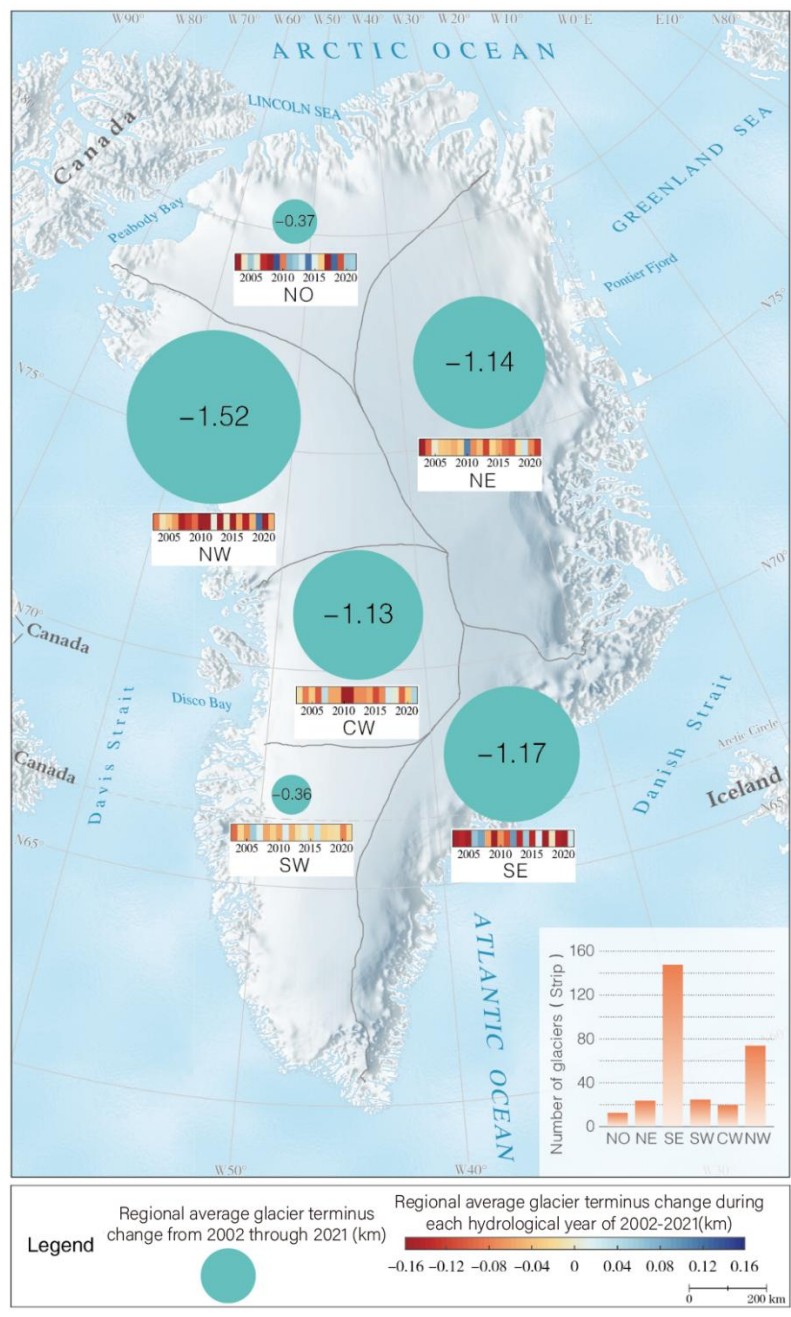

Figure 10: Regional average glacier calving front changes and glacier counts for major Greenland basins, 2002–2021. Each circle on the map is scaled to the net terminus retreat (km) of all marine-terminating glaciers within that basin over the study period; the colour bar beneath each circle shows annual calving front change for each hydrological year. The inset histogram indicates the number of glaciers per basin.

Figure 11 summarizes the temporal evolution of calving front position by basin, capturing both the magnitude and variability of annual changes. Most basins exhibited consistent retreat over the two-decade period, with pronounced



acceleration in frontal retreat occurring in the northeast and southeast since the mid-2000s. In contrast, changes in the southwest and north basins remained relatively limited throughout the record. The interannual dynamics of calving front position change was also basin-dependent. The north, southeast, and northwest basins exhibited the largest year-to-year fluctuations, with 20-year average interannual variability (AVG) values of 0.09 km, 0.12 km, and 0.10 km, and corresponding standard deviations

(STD) of 0.11 km, 0.12 km, and 0.08 km, respectively. In contrast, northeast, southwest, and west basins showed more moderate interannual variation (AVG: 0.07 km, 0.04 km, and 0.07 km; STD: 0.07 km, 0.04 km, and 0.07 km, respectively). These results highlight regional heterogeneity in glacier terminus behavior, influenced by both geometric and climatic factors (Grimes et al., 2024; Black and Joughin, 2021), and underscore the value of dense temporal sampling in resolving basin-specific dynamics.

In addition, the dataset has been successfully integrated as a time-varying boundary condition in high-resolution transient simulations of Sermeq Kujalleq using the Ice-sheet and Sea-level System Model (ISSM) for the period 2016–2022 (Lu et al., 2025). By assimilating sub-monthly calving front positions, the model is able to explain over 76% of the observed ice velocity variations, including seasonal accelerations up to 30 km upstream of the terminus. However, residual spatial and temporal velocity misfits remain, particularly near the grounding line, and are strongly correlated with fluctuations in height above

flotation within 10 km of the front. Accounting for these effects through a basal shear stress scaling approach reduces the mean velocity misfit by more than 90%, highlighting the critical role of terminus retreat–induced changes in effective pressure and basal conditions in modulating ice-flow dynamics. These results demonstrate the utility of the dataset not only for frontal change monitoring but also for improving model–data integration and enhancing the physical realism of ice flow projections under dynamic boundary conditions.

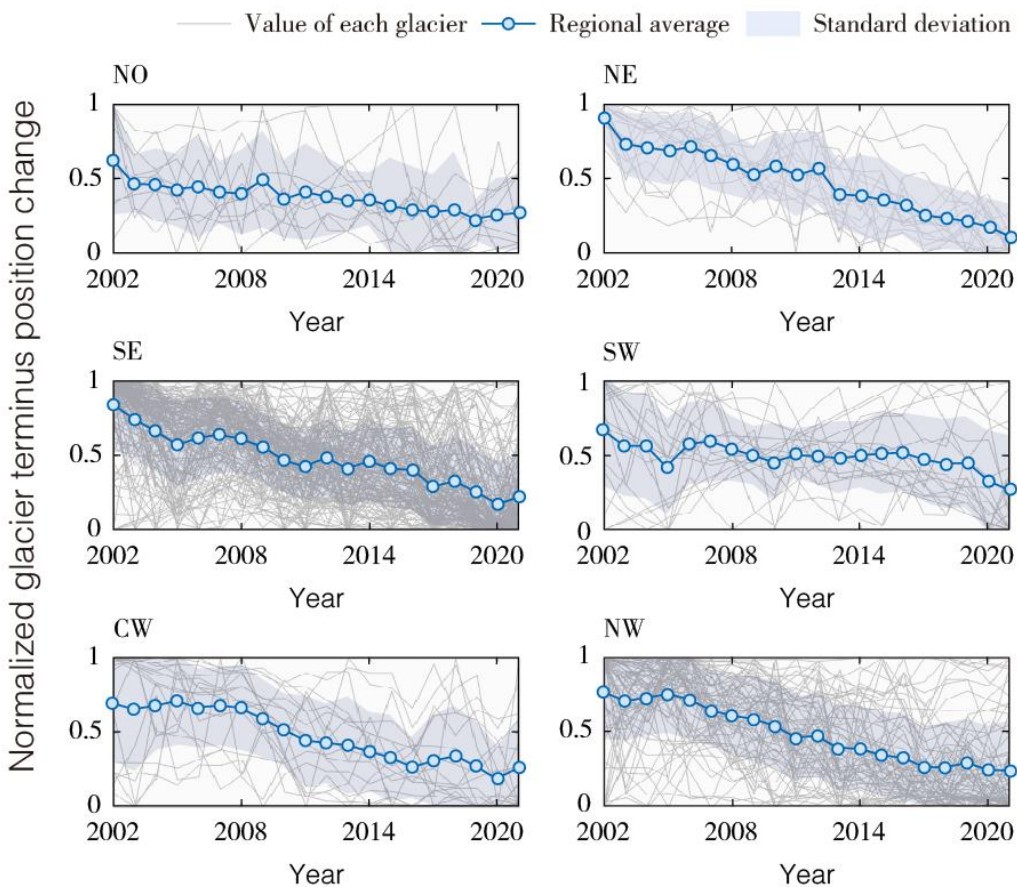

**Figure 11: Normalized terminus position time series for individual glaciers (gray lines) and basin-average calving front change (blue line) across six Greenland drainage basins from 2002 through 2021. The shaded area indicates ±1 standard deviation of the glacier-specific values around the regional mean.**

## 4 Code and data availability

The full dataset of manually delineated glacier calving front positions for ~290 outlet glaciers across Greenland from 2002 through 2021 is openly available at https://doi.org/10.5281/zenodo.16879054 (Xi et al., 2025). All calving front positions are provided in geopackage format, organized by glacier ID. Comprehensive metadata is included to support filtering by date, sensor platform, and delineation tool. The delineation was performed using the open-source GEEDiT and ArcGIS. Supplementary high-resolution reference data used for validation are accessible via their respective platforms (subject to

license or availability).



## 5 Conclusions

Accurate, high-resolution records of glacier calving front positions are essential for understanding outlet glacier dynamics, quantifying mass loss, and constraining ice sheet models. This study presents a spatially extensive and seasonally resolved dataset of calving front positions for ~290 outlet glaciers across Greenland between 2002 and 2021. Based on manual delineation from multi-source satellite imagery and supported by standardized workflows, the dataset contains approximately 12,000 calving front positions. Validation against high-resolution PlanetScope imagery and established datasets (TermPicks, AutoTerm, MEaSUREs and CALFIN) shows that the positional accuracy of our manually digitized fronts ranges from ~40 to 100 m, with sub-10 m agreement in some scenes. The dataset demonstrates enhanced performance under challenging conditions—such as mélange cover or complex frontal geometries—where automated algorithms often underperform. In such contexts, human interpretation ensures better fidelity to physical glacier boundaries and preserves fine-scale morphological detail.

Beyond serving as an observational benchmark, the dataset has proven effective in ice-flow modeling applications, where time-varying calving front positions help constrain boundary conditions and improve the realism of transient simulations. Regionally, the dataset reveals substantial spatial heterogeneity in glacier retreat, with stronger multi-year retreat and variability observed in the southeast, northeast, and northwest sectors of the ice sheet.

Overall, this contribution provides a high-quality reference for glaciological studies and algorithm development. It fills critical gaps in existing records and is openly available to support future research on calving dynamics, machine learning–based extraction, and model–data integration across the Greenland Ice Sheet.

## Author Contributions

XL produced, managed, and analyzed the dataset and wrote the manuscript. LJ provided academic supervision and conceptual guidance throughout the study, and contributed to the revision and refinement of the manuscript. DL and YL assisted with dataset generation and analysis, and supported the data processing workflow. AS and SJL contributed to scientific discussions, provided insights into glaciological processes and data interpretation, and offered critical feedback that improved the manuscript.

## Competing interests

The contact author has declared that none of the authors has any competing interests.





**Acknowledgments**

This work was funded by the National Key R&D Program of China (Grant Nos. 2018YFC1406102 and 2017YFA0603103), the Strategic Priority Research Program of the Chinese Academy of Sciences (Grant No. XDA19070104).

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
