# Peer review of "Calving front positions for Greenland outlet glaciers (2002–2021): a spatially extensive seasonal record and benchmark dataset for algorithm validation"

_Earth System Science Data, 2025_

## Referee Comment (RC1)

**General comments**

The authors describe the creation of a dataset of terminus positions from multi-source satellite imagery for ~290 marine-terminating glaciers containing ~12,000 individual traces. The terminus positions have been manually delineated, and I commence the authors for the effort that goes into that. However, I fail to see the novelty of the dataset. The authors base the majority of terminus delineations on Landsat and Sentinel imagery and use existing methods such as GEEDiT, so that I find it difficult to believe that the dataset presented here has a higher accuracy than currently existing datasets. The presented dataset contains a variety of glaciers (land-, lake-, marine-terminating) so that a) the statement that the dataset contains delineations for 290 marine-terminating glaciers is incorrect, and b) the analysis of the data is skewed and needs to be revised. There are inconsistencies and a lack of detail throughout the methodology section that make it difficult to assess how the underlying satellite was processed. The validation of the data is insufficient with one of the two methods applied not being available for the majority of the time period that the dataset covers, and the other being a comparison to existing data that does not provide a measure of accuracy of the dataset presented.

Please find my more detailed comments below:

**Major**

1. The manuscript switches between the terms "marine-terminating outlet glaciers" and "outlet glaciers" making it unclear what the focus of this study is. When comparing the dataset presented to other existing datasets, the terminology makes a big difference as e.g. TermPicks (Goliber and Black, 2021; Goliber et al., 2022) contains only delineations for marine-terminating glaciers.

2. The validation of the terminus traces needs to be improved significantly.
    a. Currently, visual validation is carried out using PlanetScope imagery which is only available from 2014 onwards and can therefore not be applied to the majority of the dataset. The authors show only 6 examples of how their delineations match with PlanetScope imagery and present no further quantification of errors. This is insufficient as a validation method.
    b.  A plausibility check is mentioned but not explained any further,
    c. The only quantifiable validation is presented by an Average Minimum Distance (AMD) which is calculated between this dataset and TermPicks and/or MEaSUREs data. The authors do not explain which delineation is used as reference line (TermPicks or MEaSUREs) and present a single value as validation. The AMD also does not provide a measure of accuracy but rather states the difference between two delineations with no indication as to which one is more accurate.

3. The authors state that panchromatically sharpened Landsat-5 imagery is used to delineate glacier fronts, which is not possible as Landsat-5 does not have a panchromatic band. In addition, there is little detail on the processing of satellite imagery outside of GEEDiT (e.g. terrain corrections, filtering). The limited detail and inconsistencies in the description of the methods make it difficult to trust the resulting data.

4. The dataset is not consistent and needs thorough revisions. In its current state, the dataset contains five different types of glaciers:
   a. Marine terminating glaciers (227)
   b. Land-terminating glaciers (21)
   c. Lake terminating glacier (5)
   d. Peripheral glaciers (22)
   e. Others including ice that I would not classify as a glacier (6), sections that are part of a larger glacier (1) or double delineations (3).

   While the majority of the data is on marine-terminating glaciers, the inclusion of other glaciers skews the data and makes the presented analysis (Figures 10, 11) redundant. It should be ensured that the dataset matches the scope of the study i.e. if the study is aimed at marine-terminating glaciers, the dataset should only contain marine-terminating glaciers; if the study is aimed at outlet glaciers, the dataset should include all lake- and land-terminating glaciers. It should be made clear whether the dataset is focused on glaciers that are directly connected to the Greenland Ice Sheet or also includes peripheral glaciers. The authors further state the dataset contains comprehensive metadata, yet it contains less than comparable datasets such as TermPicks (Goliber and Black, 2021).

5. The authors state that the presented dataset has a larger spatial coverage than existing dataset, which is misleading as it is only true when including non-marine-terminating glaciers.

6. There is no consistency in glacier names throughout the manuscript with the example glaciers being spelled wrong and differently numerous times. I suggest picking either the official name or Greenlandic name for the example glaciers and use it consistently throughout the manuscript.

**MINOR**

**Abstract**

Line 19-20: Maybe rephrase. It is not quite clear for what application it would be insufficient detail for. Also, did you test the performance and transferability of current datasets in automated calving front detection? If not, I would leave this out.

Line 21: I fail to see how your dataset can be high accuracy (or higher than other datasets) given that you use Landsat and Sentinel imagery for the majority of the delineations. Similarly, when comparing your data to TermPicks I cannot see a higher spatial extent than other datasets. Consider omitting this sentence or rephrasing.

Line 23: Why do you say approximately 12,000 traces for approximately 290 glaciers. You should state the accurate numbers, which are easily determined.

Line 28: It is unclear what the offsets of 40-100m relate to in this context. Consider rephrasing.

**Introduction**

Line 37-40: I would combine these two sentences e.g. Mass loss from the Greenland Ice Sheet contributes significantly to global sea level rise (Sheperd et al., 2018; Frederikse et al., 2020), with nearly half of this dynamic ice loss being attributed to frontal ablation (Enderlin et al., 2014; Mouginot et al., 2019).

Line 44: Consider adding references for Fahrner et al., (2025) and KC et al., (2025) as these publications address frontal ablation over large spatio-temporal scales.

Line 47: Consider adding Fahrner et al., (2021) to the reference list as you do refer to it later on.

Line 66: "…marine-terminating glaciers show highly non-linear behaviour "– While this is true for individual glaciers, many studies have shown that their behaviour on a regional scale is linear (e.g. Cowton et al., 2018; Fahrner et al., 2021). Consider rephrasing.

Figure 1: Please use the full names of the glaciers (e.g. Petermann Glacier, Jakobshavn Isbræ), check the spelling (e.g. Isstrøm) and be consistent (Gletscher vs Glacier). Panels should be labelled from A) overview map to G) Kangilinguata Sermia. Also "Sermia" should be spelled with a captial "S". This glacier is also not marine-terminating as is clearly visible in the image which contradicts previous statements that the dataset is looking at marine-terminating glaciers.

**Data and Methodology**

Line 108: GEEDiT is the acronym for the "Google Earth Engine Digitisation Tool" not the Glacier Extraction and Evaluation Dataset Tool as stated (Lea, 2018).

Line 113 - 114: How does the use of these auxiliary images impact delineation uncertainties - e.g. how can you be certain that the calving front has not changed within ±15 days?

Line 120: Be consistent with the naming of the satellites e.g. most satellites are introduced by the written-out name followed by the abbreviation in parenthesis except for ERS1/2.

Table 2: Landsat-5 does not have a panchromatic band so it cannot be resampled to a 15 m resolution.

Line 133: "…quality control and metadata creation" replace "and" with ","

Line 136 – 137: Repetition from line 125-126. Remove

Line 137: " MODIS OD09GQ?" - Is that specific MODIS imagery and therefore different from the other MODIS imagery you are using? Be consistent with the naming of data - if this is the MODIS data you use throughout, stick with the actual name. Also, auxiliary dataset for what? MODIS data?

Line 138: Is the NDWI only calculated for MODIS?

Line 141-142:  I struggle to see how an image with a resolution of 250m provides a reliable source for detecting a calving front even as a reference. Maybe provide more detail here.

Line 142-143: "ENVISAT ASAR (Level-1B) and ERS-1/2 SAR (Level-1.5) scenes were downloaded from the ESA archives" can be removed as you already state prior that the data was downloaded from ESA.

Line 143-144: Be more specific regarding the calibration and terrain correction, e.g., what DEM was used for the terrain correction?

Line 145: Same as above, what DEM was used and what filter applied.

Line 146: Why was the data not reprojected to EPSG:3413 Polar Stereographic? This is the common format for datasets that cover Greenland.

Line 148-149: These sentences contradict each other as you first say that delineations were done on native-resolution imagery and then say that you used pan-sharpened images.

Line 150-151: You say in Line 113-114 that this was only done for Landsat-7, but now it reads like it was done for other satellites too?

Line 152: How did you manually adjust the delineations? Did you use a different image or the same image? Be more specific.

Line 153: Please explain what the plausibility check entails as this is the only mention of it.

Figure 2: The figure needs to be redone as it is currently confusing. Under preprocessing you mention everything that has been done to the satellite imagery, but you don't specify what was done to which imagery. It is somewhat clear that the imagery on the left corresponds with the boxes, but I think it could be made clearer.

I don't think the "Sets parameter.." is necessary unless you specify which parameters.

I am still confused by the editing and modification of the delineations.

The QA process is not clearly explained at all and needs additional details.

Also, again check the glacier names for consistency, spelling etc.

Line 162: You mention uncertainties in delineation of 25m, however Brough et al., (2019) and (Fahrner et al., (2021) contradict this statement as they performed error checks for Landsat 4 (30 m) and Landsat 8 (15 m) and came to the conclusion that the error is much less than the pixel resolution.

Line 165-166: Planet imagery is only available from 2014, so there is no data for over half of the time period that your dataset covers.  I fail to see how this can be a quality check, especially since it is also only a visual validation. I suggest the authors should find a way to quantify the uncertainties.

Line 175: "Its fine spatial detail and temporal coverage make it a valuable benchmark for evaluating glacier front delineations." That might be correct, but it is only available from 2014 onwards so that it is not useful for longer time series such as your dataset.

Line 177: Glacier names are incomplete and spelled wrong.

Line 183: "...capturing key features with sub-pixel accuracy..." if I understand this correctly you say that your delineations capture sub-pixel features. If so, I think that might just be coincidence as I don't think you'd be able to accurately delineate features within a pixel. Otherwise, please rephrase.

Line 188-189: I don't think the fact that this glacier is included in this dataset but not in others warrants the claim that this dataset has a greater spatial coverage. Especially since some of the other datasets do not contain land-terminating glaciers such as Kangilinguata Sermia. Consider rephrasing.

Line 201: Double space between "Kangilinguata" and "glacier".

Line 201: "...which is not included in any of the existing datasets,..." see previous comment on Line 188-189.

Line 207: Rephrase. I think you're trying to say that your dataset outperforms automated approaches but not all manually delineated datasets.

Line 227: Make sure the citation is correct – You are citing Goliber et al., (2022) which is the publication, but you should be citing the dataset Goliber and Black, 2021)

Line 226 – 228: I struggle to see how a close alignment to existing datasets is showing that your dataset is novel or more accurate. Maybe rephrase?

Line 236: It is unclear here if TermPicks of MEaSUREs is used as reference line. I assume that the AMD was calculated for bother and only the maximum is mentioned? Please be more specific.

Line 279-280: Replace "high positional consistency with" with "corresponds well with"

Line 280-281: "... depending on glacier geometry, image quality, and surface contrast." I would leave this out as you do not provide any measure on how any of these affect your accuracy.

Line 281: This is the first time semi-automated methods are mentioned. Unless it is introduced prior with a specific reference, I would leave it out.

Line 286 – 287: From the figures in the manuscript, I struggle to see the increased spatial and temporal coverage compared to TermPicks. If anything, the dataset presented here seems to have less temporal coverage than existing datasets (e.g. Figure 7).

Line 287 – 288: Given that the dataset contains a multitude of glaciers that are land- and lake-terminating and has no consistent temporal resolution, I would remove this sentence. Currently the dataset contains 227 marine-terminating glaciers, and the temporal resolution varies from annually to monthly.

Line 289 -290: It states here that there is "high-precision agreement with PlanetScope imagery" however earlier it is stated that this was only accessed visually. There is no proof in the manuscript that would support the claim of "positional discrepancies are consistently below 10m". I suggest the authors rephrase or provide a detailed description of how this was determined.

**Data product and usage notes:**

Line 301: The name of the individual files "Calving_front_positions_for_Greenland_outlet_glaciers_2002_2021 — New_NeighborGID115" is unnecessarily long. I would recommend to either just use the

Glacier ID or use the glacier name. In the current format, the key information (GlacierID) is almost hidden and I have to use half my screen in QGIS to see it.

Line 308: The metadata should contain at least the official glacier name as specified in TermPicks or Bjork et al., (2015).

Line 313: The "season" field can be left out as it is quite clear from the "Date" field what season the delineation was taken in.

Line 318: Check the glacier names for spelling and consistency.

Line 344: Write these values as values±STD km.

Line 350 – 351: If you only model one glacier it's not really the dataset that you are using but a subset. I would leave this sentence out as it doesn't really add anything.

**Conclusions**

Line 376: It should be mentioned here that the validation against PlanetScope imagery was visual only, and that the AMD was only determined for TermPicks and/or (?) MEaSUREs. Currently it reads like the validation of the dataset was extensive, which unfortunately, it was not.

**References**

Brough, S., Carr, J.R., Ross, N. and Lea, J.M., (2019) Exceptional retreat of Kangerlussuaq Glacier, east Greenland, between 2016 and 2018. *Frontiers in Earth Science*, 7123.

Cowton, T.R., Sole, A.J., Nienow, P.W., Slater, D.A. and Christoffersen, P., (2018) Linear response of east Greenland's tidewater glaciers to ocean/atmosphere warming. *Proceedings of the National Academy of Sciences*, 11531, pp.7907–7912.

Fahrner, D., Lea, J.M., Brough, S., Mair, D.W.F. and Abermann, J., (2021) Linear response of the Greenland ice sheet's tidewater glacier terminus positions to climate. *Journal of Glaciology*, [online] 67262, pp.193–203. Available at: https://www.cambridge.org/core/article/linear-response-of-the-greenland-ice-sheets-tidewater-glacier-terminus-positions-to-climate/6B3723E3A0E94012A1DB9D3E49246AF2.

Fahrner, D., Slater, D.A., KC, A., Cenedese, C., Sutherland, D.A., Enderlin, E., de Jong, M.F., Kjeldsen, K.K., Wood, M., Nienow, P., Nowicki, S. and Wagner, T.J.W., (2025) A Frontal Ablation Dataset for 49 Tidewater Glaciers in Greenland. *Nature Scientific Data*.

Goliber, S. and Black, T., (2021) *TermPicks: A century of Greenland glacier terminus data for use inmachine learning applications (Version 1)*. Available at: https://zenodo.org/records/5117931.

Goliber, S., Black, T., Catania, G., Lea, J.M., Olsen, H., Cheng, D., Bevan, S., Bjørk, A., Bunce, C., Brough, S., Carr, J.R., Cowton, T., Gardner, A., Fahrner, D., Hill, E., Joughin, I., Korsgaard, N.J., Luckman, A., Moon, T., Murray, T., Sole, A., Wood, M. and Zhang, E., (2022) TermPicks: a century of Greenland glacier terminus data for use in scientific and machine learning applications. *Cryosphere*, [online] 168, pp.3215–3233. Available at: https://zenodo.org/record/5117931 [Accessed 27 May 2022].

KC, A., Enderlin, E.M., Fahrner, D., Moon, T. and Carroll, D., (2025) Seasonality in terminus ablation rates for the glaciers in  Greenland (Kalaallit Nunaat). *The Cryosphere*, [online] 198, pp.3089–3106. Available at: https://tc.copernicus.org/articles/19/3089/2025/.

Lea, J.M., (2018) The Google Earth Engine Digitisation Tool (GEEDiT) and the Margin change Quantification Tool (MaQiT)--simple tools for the rapid mapping and quantification of changing Earth surface margins. *Earth Surface Dynamics*, 63, pp.551–561.

---

## Author Comment (AC1)

**Response to review**

Manuscript ID: ESSD-2025-304

**Title:** Calving front positions for Greenland outlet glaciers (2002–2021): a spatially extensive seasonal record and benchmark dataset for algorithm validation

Dear reviewer,

We thank the reviewer for the careful and detailed evaluation. The original submission did not clearly demonstrate the dataset's scope and reliability, and we are undertaking substantial revisions to address all points raised. Your constructive suggestions have provided clear guidance, and we have revised the manuscript and dataset to further clarify their quality, coverage, and applicability.

- 1. Clarified dataset coverage and scope by explicitly including marine-, land-, lake-terminating, and floating-tongue glaciers; expanded digitization to improve spatial completeness; and added *GlacierName* and *GlacierType* fields to the dataset.
- 2. Implemented a comprehensive cross-dataset Average Minimum Distance (AMD) assessment by identifying or interpolating at least one overlapping delineation per glacier and comparing our dataset with TermPicks, AutoTerm, CALFIN, and Black & Joughin (2023).
- 3. Revised overly assertive wording, clarified the relevant limitations, and refined and expanded the method section.

We hope that the revisions fully address all of your comments, and we appreciate the detailed critique, which has significantly improved the clarity and quality of both the manuscript and the dataset.

Best regards, Xi Lu and all co-authors

**General comments**

The authors describe the creation of a dataset of terminus positions from multi-source satellite imagery for ~290 marine-terminating glaciers containing ~12,000 individual traces. The terminus positions have been manually delineated, and I commence the authors for the effort that goes into that. However, I fail to see the novelty of the dataset.

The authors base the majority of terminus delineations on Landsat and Sentinel imagery and use existing methods such as GEEDiT, so that I find it difficult to believe that the dataset presented here has a higher accuracy than currently existing datasets. The presented dataset contains a variety of glaciers (land-, lake-, marine-terminating) so that a) the statement that the dataset contains delineations for 290 marine-terminating glaciers is incorrect, and b) the analysis of the data is skewed and needs to be revised. There are inconsistencies and a lack of detail throughout the methodology section that make it difficult to assess how the underlying satellite was processed. The validation of the data is

insufficient with one of the two methods applied not being available for the majority of the time period that the dataset covers, and the other being a comparison to existing data that does not provide a measure of accuracy of the dataset presented.

Response: We thank the reviewer for this thoughtful assessment. We agree that the dataset itself does not introduce methodological novelty, as it employs established tools such as GEEDiT and existing satellite archives. However, while recent novel deep-learning approaches have achieved notable progress, they remain highly dependent on large, reliable manual datasets for training and validation (e.g., Cheng et al., 2021; Goliber et al., 2022). Such manually curated data remain indispensable, particularly in complex terminus environments where automated methods can exhibit local errors of 1-2 km. In addition, the TermPicks compilation, which aggregates previously published manually delineated fronts, focuses solely on marine-terminating glaciers and lacks consistent seasonal resolution due to the heterogeneity of its source studies. For example, only 221 glaciers in TermPicks have seasonal resolution, and coverage varies by glacier and through time. Therefore, although not algorithmically innovative, our dataset provides a critical dataset for training, validation, and benchmarking, complementing ongoing advances in automated calving-front detection and supplementing existing manual datasets. The main contribution of this work lies in the creation of a spatially extensive, long-term, high-precision, and manually-verified dataset covering multiple glacier types, which offers essential reference material for both glaciological analyses and machine-learning development.

Regarding the dataset's scope and glacier classification, our initial goal—beyond supporting automated calving-front detection—was to create a dataset suitable for large-scale analyses of the seasonal behavior of Greenland outlet glaciers, such as joint studies with seasonal velocity products derived from Landsat imagery. Accordingly, we used the spatial extent of glaciers covered by Landsat-based seasonal velocity fields as the selection reference (Fig. 1; Rosenau et al. (2015)). Glaciers outside this extent were assumed to show minimal frontal change and were therefore excluded. Within this framework, the dataset effectively covers all glaciers exhibiting detectable seasonal velocity variations. Because marine-terminating glaciers constitute the majority of these outlets, the initial version erroneously referred to them collectively as "marine-terminating glaciers" without explicitly distinguishing glacier types.

Considering the availability of existing datasets and your suggestions, we have now refined the definition of coverage to explicitly include marine-, land-, lake-terminating and floating-tongue glaciers, rather than restricting it to those defined by long-term seasonal velocity coverage. The dataset is being extended to incorporate additional glaciers of each type, which will be clearly labelled in the metadata including GlacierType and GlacierName fields to enhance accessibility and usability for the wider cryosphere community. The additional glacier-front delineations were identified using a combination of existing inventories map from NunaGIS, glacier name from Bjørk et al. (2015), icemarginal lake maps (Dømgaard et al. (2024)), and satellite imagery. Peripheral glaciers with negligible dynamic activity will be retained but not expanded.

Fig. 9. 302 locations (red circles) and names of all analysed outlet glaciers. The official spelling of the glacier names was chosen according to NunaGIS (2015). Unofficial glacier names are marked italic. The grey flow lines are determined by Gebler (2009). The hill-shading was performed with the elevation data from Ekholm (1996). The ice divide is highlighted with the red dotted line. The dashed blue lines indicate the regional boundaries.

**Fig.1** Selection basis of the research area, defined by the spatial extent of outlet glaciers covered by Landsat-derived seasonal velocity fields (Rosenau et al., 2015).

To address concerns regarding validation and data reliability, we agree that the initial evaluation—based on six representative glaciers—was not sufficient to capture dataset-wide variability, mainly due to the difficulty of finding overlapping delineations for the same glacier on the same date. As also noted in response to another reviewer, we are now conducting an expanded validation that systematically compares our dataset with four major products (TermPicks, AutoTerm, CALFIN, and Black & Joughin, 2023) using Average Minimum Distance (AMD) metrics. To overcome the challenge of missing sameday overlaps, we apply interpolation between available records from existing datasets to improve temporal matching. For each overlapping case, we calculate AMD statistics and summarize the results by glacier and by region (mean and median).

The methodology section is being revised to include detailed descriptions of satellite data preprocessing, delineation procedures, and validation steps. It will also clarify glacier selection criteria, specify the sensors and temporal intervals used, and explicitly distinguish between PlanetScope-based validation and inter-dataset comparisons.

In summary, while this dataset does not introduce methodological innovations, its value lies in providing the most spatially extensive, seasonally resolved, and manually verified record of calving front positions across Greenland. The forthcoming revised version will incorporate additional glacier types, extended validation, and detailed methodological clarification to strengthen its robustness and community relevance.

**Response to major comments**

1. The manuscript switches between the terms "marine-terminating outlet glaciers" and "outlet glaciers" making it unclear what the focus of this study is. When comparing the dataset presented to other existing datasets, the terminology makes a big difference as e.g. TermPicks (Goliber and Black, 2021; Goliber et al., 2022) contains only delineations for marine-terminating glaciers.

Response: Thank you for pointing this out. The dataset was initially defined by the extent of outlet glaciers with detectable seasonal velocity variations, without explicit distinction by glacier type. As marine-terminating glaciers make up the majority, the manuscript at times erroneously used "outlet glaciers" and "marine-terminating glaciers" interchangeably.

Following your suggestion, we are expanding the dataset to include more marine-, lake- and land-terminating glaciers, which are underrepresented in existing datasets but increasingly important for recent studies. In the revised version, we will use the general term "glaciers" for the dataset overall, while retaining type-specific terms where relevant. All terminology has been standardized, and glacier-type classification will be clearly defined in the methods. Each record now includes a GlacierType attribute in the metadata for clarity.

- 2. The validation of the terminus traces needs to be improved significantly.
  - **a.** Currently, visual validation is carried out using PlanetScope imagery which is only available from 2014 onwards and can therefore not be applied to the majority of the dataset. The authors show only 6 examples of how their delineations match with PlanetScope imagery and present no further quantification of errors. This is insufficient as a validation method.
    - **b.** A plausibility check is mentioned but not explained any further.
  - **c.** The only quantifiable validation is presented by an Average Minimum Distance (AMD) which is calculated between this dataset and TermPicks and/or MEaSUREs data. The authors do not explain which delineation is used as reference line (TermPicks or MEaSUREs) and present a single value as validation. The AMD

also does not provide a measure of accuracy but rather states the difference between two delineations with no indication as to which one is more accurate.

*Response*: We fully agree that validation is a critical component of this dataset and appreciate the reviewer's detailed comments.

**(a) PlanetScope-based validation:**

PlanetScope imagery (3 m resolution) was used for qualitative validation where such data are available (2014–2021). The initial purpose was to visually confirm the geometric fidelity of the delineated ice—ocean boundary, particularly in complex environments such as mélange-filled fjords, heavily shadowed termini, or regions with strong illumination gradients. Because the availability of such high-resolution imagery is limited, six representative glaciers from different sectors of Greenland were selected to illustrate how the manual delineations correspond to fine-scale features visible in the optical data.

In the revised manuscript, PlanetScope scenes are used not only for visual assessment but also for multiple repeated measurements of calving fronts. For these cases, AMD values are computed directly to provide a quantitative estimate of the precision of the repeated manual delineations and, where available, to support quantitative comparisons with contemporaneous entries from other datasets. We acknowledge that the restricted temporal availability of PlanetScope imagery does not represent the full 2002–2021 period. A broader quantitative cross-dataset validation (detailed below) is implemented to achieve comprehensive spatial coverage across the entire dataset, extending beyond the limited temporal window where PlanetScope observations exist.

**(b) Plausibility check:**

This step involves a post-processing consistency check, including a visual review of delineations based on time series of calving frontal changes extracted along each glacier's central flowline. Inconsistent fronts—such as those exhibiting abrupt, temporally unrealistic jumps—are removed based on this temporal consistency assessment. We will elaborate on this procedure in the revised methodology section and provide an example to illustrate its implementation.

**(c) Quantitative validation and AMD metrics:**

In the initial submission, AMD values were calculated between our dataset and existing manual reference datasets (TermPicks or MEaSUREs) for six glaciers. Manual delineations are generally regarded as the most reliable reference, as they are based on visual interpretation of remote sensing imagery on a scene-by-scene basis. Consequently, automated products are typically validated against manual datasets, while manual compilations, such as TermPicks, assess internal consistency through comparisons of overlapping delineations from different authors (e.g., Goliber et al., 2022; TermPicks, median error = 107 m, range = 58.6–7,350 m). As a new manually delineated dataset, our product was initially validated quantitatively against overlapping entries from manual datasets (e.g., TermPicks) and qualitatively using both PlanetScope imagery and automated products to assess geometric fidelity, highlight the limitations of existing methods, and demonstrate the potential contribution of our dataset. This intercomparison-based strategy aligns with standard validation approaches used in existing calving-front products.

To further enhance reliability, we are now conducting an expanded cross-dataset validation against TermPicks, AutoTerm, CALFIN, and Black & Joughin (2023) using AMD statistics. Because exact same-day overlaps are rare (e.g., TermPicks manually traces only ~5.8 % of available Landsat scenes per glacier), we apply temporal interpolation between adjacent delineations in existing datasets to obtain near-coincident comparisons. Each comparison will include mean and median of AMD values, summarized by glacier and region, providing a transparent and quantitative measure of inter-dataset differences. Together, these enhancements establish a comprehensive validation framework combining high-resolution qualitative checks, inter-dataset quantitative analysis, and internal consistency screening. The full details will be provided in the revised methodology and validation sections.

- 3. The authors state that panchromatically sharpened Landsat-5 imagery is used to delineate glacier fronts, which is not possible as Landsat-5 does not have a panchromatic band. In addition, there is little detail on the processing of satellite imagery outside of GEEDiT (e.g. terrain corrections, filtering). The limited detail and inconsistencies in the description of the methods make it difficult to trust the resulting data.
  - Response: Thank you for pointing this out. This error was due to a table-editing oversight where the Landsat-7 entry was inadvertently duplicated. The text and table have been corrected to specify that panchromatic images apply only to Landsat-7 and 8. Additional details on image preprocessing—including radiometric and terrain correction, cloud and shadow masking, and contrast enhancement—will be added to ensure methodological transparency and consistency across all image sources.
- 4. The dataset is not consistent and needs thorough revisions. In its current state, the dataset contains five different types of glaciers:
  - a. Marine terminating glaciers (227)
  - b. Land-terminating glaciers (21)
  - c. Lake terminating glacier (5)
  - d. Peripheral glaciers (22)
  - e. Others including ice that I would not classify as a glacier (6), sections that are part of a larger glacier (1) or double delineations (3).

While the majority of the data is on marine-terminating glaciers, the inclusion of other glaciers skews the data and makes the presented analysis (Figures 10, 11) redundant. It should be ensured that the dataset matches the scope of the study i.e. if the study is aimed at marine-terminating glaciers, the dataset should only contain marine-terminating glaciers; if the study is aimed at outlet glaciers, the dataset should include all lake- and land-terminating glaciers. It should be made clear whether the dataset is focused on glaciers that are directly connected to the Greenland Ice Sheet or also includes peripheral glaciers. The authors further state the dataset contains comprehensive metadata, yet it contains less than comparable datasets such as TermPicks (Goliber and Black, 2021).

Response: Thank you for this valuable comment. As noted in our general response, the dataset's initial scope—beyond supporting automated calving-front detection—was designed to enable large-scale digitization of the seasonal calving front behavior of Greenland outlet glaciers. Accordingly, we used the spatial extent of glaciers covered by Landsat-based seasonal velocity fields as the selection reference (Fig. 1; Rosenau et al., 2015). Glaciers outside this extent were assumed to exhibit minimal frontal change and were therefore excluded. Within this framework, the dataset effectively includes all glaciers with detectable seasonal velocity variations without explicitly distinguishing glacier types.

Following your suggestion and considering the coverage of existing datasets, we have refined the definition of coverage to explicitly include marine-, land-, lake-terminating, and floating-tongue glaciers, rather than limiting it to velocity-defined outlets. The dataset is being extended to incorporate additional glaciers of each type, clearly labelled in the metadata to improve accessibility and usability for the cryosphere community.

- 5. The authors state that the presented dataset has a larger spatial coverage than existing dataset, which is misleading as it is only true when including non-marine-terminating glaciers.
  - Response: Thank you for this clarification. The original statement on spatial coverage did not distinguish glacier types and was based on the total number of delineated glaciers (~290 outlet glaciers), which—even without classification—already exceeds the coverage of existing products. As noted above, we are refining the dataset's classification and expanding it to include clearly defined marine-, land-, lake-terminating, and floating-tongue glaciers. Once the updated dataset is completed, the total number of glaciers will further increase. Nevertheless, in the revised manuscript, we will moderate this statement to avoid overly strong claims.
- 6. There is no consistency in glacier names throughout the manuscript with the example glaciers being spelled wrong and differently numerous times. I suggest picking either the official name or Greenlandic name for the example glaciers and use it consistently throughout the manuscript
  - *Response*: All glacier names will be corrected and standardized to their official or Greenlandic forms based on Bjørk et al. (2015) and map from NunaGIS to ensure consistency throughout the manuscript and figures.

**Response to minor comments**

1. L19-20 -Maybe rephrase. It is not quite clear for what application it would be insufficient detail for. Also, did you test the performance and transferability of current datasets in automated calving front detection? If not, I would leave this out.

Response: Thank you for this suggestion. Our intention was to highlight that existing datasets provide limited glacier coverage and temporal sampling, which constrains their usefulness for training, validating, and assessing the generalization ability of automated detection algorithms. To avoid misunderstanding, we have rephrased the sentence to clarify that the limitation refers to spatial and temporal

consistency rather than any performance testing.

**Text from the revised version (L19-L20)**

Existing manual calving-front datasets vary in spatial and temporal coverage, limiting their ability to support the training, validation, and generalization of automated detection approaches.

2. L21 –I fail to see how your dataset can be high accuracy (or higher than other datasets) given that you use Landsat and Sentinel imagery for the majority of the delineations. Similarly, when comparing your data to TermPicks I cannot see a higher spatial extent than other datasets. Consider omitting this sentence or rephrasing.

Response: As noted above, the reference to "high accuracy" was intended in comparison with automated products, rather than other manually derived datasets. From the six representative glaciers evaluated in the initial submission, our manually delineated fronts consistently performed better in complex regions—particularly along lateral margins—where automated methods often show large local errors. The earlier statement on spatial coverage referred to the total number of glaciers represented in the dataset. We will clarify this wording in the revised manuscript, especially as the ongoing expansion effort aims to include additional marine-terminating as well as other glacier types.

Given that these advantages are context-dependent, we have revised the sentence in the updated manuscript to more accurately describe the dataset's strengths and limitation.

**Text from the revised version (L21)**

This dataset provides spatially extensive, seasonally resolved, and internally consistent delineations across multiple glacier types, complementing existing calving-front records.

- 3. **L23** –Why do you say approximately 12,000 traces for approximately 290 glaciers. You should state the accurate numbers, which are easily determined. *Response*: Thank you for the suggestion. The approximation is due to the fact that some very close-by glaciers were digitized in the same file. The exact numbers will be reported in the revised manuscript once the ongoing dataset expansion is completed.
- **4.** L28 –It is unclear what the offsets of 40-100m relate to in this context. Consider rephrasing.

*Response*: The offsets of 40–100 m refers to the typical positional differences between our manually delineated fronts and reference manually datasets (TermPicks and MEaSUREs) at overlapping dates. This has been clarified in the revised text.

**5.** L37–40 –I would combine these two sentences e.g. Mass loss from the Greenland Ice Sheet contributes significantly to global sea level rise (Sheperd et al., 2018; Frederikse et al., 2020), with nearly half of this dynamic ice loss being attributed to frontal ablation (Enderlin et al., 2014; Mouginot et al., 2019).

*Response*: The two sentences have been combined and rephrased for conciseness as follows:

**Text from the revised version (L37-L40)**

Mass loss from the Greenland Ice Sheet contributes significantly to global sea-level rise (Shepherd et al., 2018; Frederikse et al., 2020), with nearly half of this dynamic ice loss attributed to frontal ablation (Enderlin et al., 2014; Mouginot et al., 2019).

6. L44 & L47 –Consider adding references for Fahrner et al., (2025) and KC et al., (2025) as these publications address frontal ablation over large spatio-temporal scales. Consider adding Fahrner et al., (2021) to the reference list as you do refer to it later on.

Response: Added citations Fahrner et al. (2021, 2025) and Kc et al. (2025).

7. L66 - ... marine-terminating glaciers show highly non-linear behaviour — While this is true for individual glaciers, many studies have shown that their behaviour on a regional scale is linear (e.g. Cowton et al., 2018; Fahrner et al., 2021). Consider rephrasing.

**Response:** Thank you for this comment. The sentence has been rephrased accordingly to reflect this distinction.

**Text from the revised version (L66)**

These approaches are especially valuable for monitoring Greenland's marine-terminating glaciers, which can exhibit nonlinear behavior in response to climatic and oceanic forcing at the individual-glacier scale (Brough et al., 2023; Cowton et al., 2018; Fahrner et al., 2021).

- 8. **Figure 1** –Please use the full names of the glaciers (e.g. Petermann Glacier, Jakobshavn Isbræ), check the spelling (e.g. Isstrøm) and be consistent (Gletscher vs Glacier). Panels should be labelled from A) overview map to G) Kangilinguata Sermia. Also "Sermia" should be spelled with a captial "S". This glacier is also not marine-terminating as is clearly visible in the image which contradicts previous statements that the dataset is looking at marine-terminating glaciers.
  - **Response:** As noted previously, all glacier names will be corrected and standardized according to map from NunaGIS and Bjørk et al. (2015). In addition, since the revised dataset will include both marine- and non-marine-terminating glaciers, the related descriptions in the text will be updated to avoid implying an exclusive focus on marine-terminating glaciers.
- 9. **L108** –GEEDiT is the acronym for the "Google Earth Engine Digitisation Tool" not the Glacier Extraction and Evaluation Dataset Tool as stated (Lea, 2018). *Response:* The full name has been corrected.
- 10. L113–114 –How does the use of these auxiliary images impact delineation uncertainties e.g. how can you be certain that the calving front has not changed within  $\pm 15$  days?

**Response:** Thank you for raising this important point. The  $\pm 15$ -day temporal window was applied only when the target Landsat-7 scene was affected by Scan Line Corrector (SLC) gaps or when crucial frontal areas were partially obscured by cloud cover. In such cases, auxiliary scenes within  $\pm 15$  days were used only for visual cross-checking, not for direct delineation. The final glacier front was always

- digitized from the primary scene to maintain temporal consistency. This clarification has been added to the revised methods section, together with a note acknowledging that this approach may introduce minor additional uncertainty for glaciers experiencing rapid frontal change.
- 11. **L120** –Be consistent with the naming of the satellites e.g. most satellites are introduced by the written-out name followed by the abbreviation in parenthesis except for ERS1/2.
  - **Response:** We have standardized the naming of all satellite missions throughout the manuscript. Full names are now provided at first mention, followed by their abbreviations in parentheses.
- 12. **Table 2** –Landsat-5 does not have a panchromatic band so it cannot be resampled to a 15 m resolution.
  - **Response:** We appreciate the reviewer's correction and apologies for the oversight. The information for Landsat 5 has been corrected.
- 13. L133–137 –"...quality control and metadata creation" replace "and" with ","; Repetition from line 125-126. Remove
  - **Response:** The redundant sentence has been removed, and the phrasing has been revised to replace "and" with a comma as suggested.
- 14. L137 "MODIS OD09GQ?" Is that specific MODIS imagery and therefore different from the other MODIS imagery you are using? Be consistent with the naming of data if this is the MODIS data you use throughout, stick with the actual name. Also, auxiliary dataset for what? MODIS data?
  - Response: The MODIS product used is specifically MOD09GQ, and we have now standardized this naming throughout the manuscript for consistency. Glacier frontal changes are generally gradual and follow seasonal patterns, so consecutive delineations typically capture the overall migration trend. However, when two temporally distant delineations appeared nearly coincident, MOD09GQ imagery was used to assess whether intermediate advance—retreat cycles or local nonlinear variations occurred, ensuring that all key transitions were captured. In a few cases—where high-resolution imagery was unavailable, but the terminus exhibited predictable stable and near-linear behavior—MOD09GQ scenes were also used for delineation to maintain seasonal continuity.
- 15. **L138**–Is the NDWI only calculated for MODIS?
  - **Response:** Yes, the NDWI was calculated only for MODIS imagery. Since the MOD09GQ product provides the red and near-infrared bands at 250 m resolution, we adopted a red–NIR normalized difference index (Red–NIR NDWI) to enhance the contrast between ice and open water.
- 16. L141-142–I struggle to see how an image with a resolution of 250m provides a reliable source for detecting a calving front even as a reference. Maybe provide more detail here.
  - **Response:** As noted above, MODIS imagery was used only in a limited number of cases, and not as the primary source for delineating a calving front. These situations occurred at glaciers where consecutive high-resolution scenes (Landsat/Sentinel) showed a stable, near-linear pattern of terminus change over an extended period. In such situations, MODIS served only to support interpolation between well-constrained high-resolution positions and to maintain seasonal continuity, rather

- than to identify the front independently. All instances in which MODIS contributed to temporal interpolation have been explicitly documented in the metadata and can be filtered by users as needed.
- 17. L142-143—"ENVISAT ASAR (Level-1B) and ERS-1/2 SAR (Level-1.5) scenes were downloaded from the ESA archives" can be removed as you already state prior that the data was downloaded from ESA.

  Response: It has been removed.
- 18. **L143-144:** Be more specific regarding the calibration and terrain correction, e.g., what DEM was used for the terrain correction?
  - **Response:** Thank you for the suggestion. We have added details specifying that radiometric calibration and terrain correction were performed using the Sentinel Application Platform (SNAP) toolbox, with topographic correction based on the ArcticDEM. This information has been included in the revised Methods section to ensure transparency and reproducibility.
- 19. L145 –Same as above, what DEM was used and what filter applied. *Response:* We thank the reviewer for the comment. The revised manuscript now specifies that ArcticDEM was used and that a Refined Lee filter was applied. These details have been added to the method section for clarity.
- 20. **L146** Why was the data not reprojected to EPSG:3413 Polar Stereographic? This is the common format for datasets that cover Greenland.
  - **Response:** The dataset was not deliberately projected to a specific coordinate system; rather, EPSG:4326 (WGS84 geographic coordinates) was adopted as a common reference to unify all imagery and derived products. This ensures consistency across data sources with different native projections and facilitates straightforward use in most GIS and web-based visualization platforms.
- 21. **L148–151** –These sentences contradict each other as you first say that delineations were done on native-resolution imagery and then say that you used pan-sharpened images. For example, you say in Line 113-114 that this was only done for Landsat-7, but in Line 150-151 now it reads like it was done for other satellites too?
  - **Response:** Thank you for the comment. The methods section originally described multiple data sources separately, which may have caused confusion. To clarify, all optical imagery was delineated at its native spatial resolution, with GEEDiT providing visually enhanced true-colour composites for clearer interpretation. In a few specific cases, auxiliary images from nearby dates were used for reference: for example, optical images affected by cloud cover, and Sentinel-1 images where the ice—water boundary was indistinct. These auxiliary scenes were used only for visual cross-checking, not for direct delineation. The text has been revised accordingly.
- 22. L152–153 –How did you manually adjust the delineations? Did you use a different image or the same image? Be more specific. Please explain what the plausibility check entails as this is the only mention of it.
  - **Response:** When clear inconsistencies were observed between the current front and preceding and/or following fronts, suggesting that the delineation might not be reliable, it was rechecked and manually refined using the original image. The plausibility check refers to a post-processing step applied after extracting calving front position variation along the glacier centerline. Time series of frontal positions were examined for expected seasonal or gradual variations. If an outlier or abrupt

- change inconsistent with neighboring observations was identified, the corresponding delineation was rechecked against the original imagery to confirm or correct potential errors. This procedure ensured temporal coherence and reduced spurious fluctuations in the final dataset.
- 23. Figure 2 –The figure needs to be redone as it is currently confusing. Under preprocessing you mention everything that has been done to the satellite imagery, but you don't specify what was done to which imagery. It is somewhat clear that the imagery on the left corresponds with the boxes, but I think it could be made clearer.

I don't think the "Sets parameter." is necessary unless you specify which parameters. I am still confused by the editing and modification of the delineations.

The QA process is not clearly explained at all and needs additional details. Also, again check the glacier names for consistency, spelling etc.

**Response:** Figure 2 will be fully redesigned to improve clarity and logical consistency. The revised version will distinguish preprocessing steps for different imagery types (Landsat, Sentinel, MODIS, SAR), specify the applied parameters and operations, and adds further detail to the delineation and QA stages (see revised manuscript for more information).

- 24. L162 –You mention uncertainties in delineation of 25m, however Brough et al., (2019) and (Fahrner et al., (2021) contradict this statement as they performed error checks for Landsat 4 (30 m) and Landsat 8 (15 m) and came to the conclusion that the error is much less than the pixel resolution.
  - **Response:** The 25 m uncertainty mentioned refers to the positional deviation relative to the original satellite imagery (precision), rather than to repeated manual digitization on a single image. As noted by another reviewer, this represents a conservative estimate, since additional uncertainties can arise from cloud cover, side-wall shadows, or fragmented fronts—conditions under which errors can reach several hundred meters (e.g., TermPicks, median error = 107 m, range = 58.6–7,350 m). The studies by Brough et al. (2019) and Fahrner et al. (2021) indeed evaluated the internal consistency of manual delineation (i.e., repeated tracing of the same image by the same operator), **which measures precision rather than accuracy**. This sentence has been revised for clarity to distinguish between these error sources.
- 25. **L165–166** –Planet imagery is only available from 2014, so there is no data for over half of the time period that your dataset covers. I fail to see how this can be a quality check, especially since it is also only a visual validation. I suggest the authors should find a way to quantify the uncertainties.
  - **Response:** As noted, PlanetScope imagery (available only after 2014) was used solely for qualitative visual assessment to illustrate the geometric reliability of the manual delineations, rather than as a dataset-wide validation tool. In the revised manuscript, PlanetScope scenes are additionally used for multiple repeated measurements of fronts, where AMD values are computed directly to provide a quantitative estimate of manual delineation precision and, where available, are quantitatively compared against contemporaneous entries from other datasets. To provide a more comprehensive and quantitative evaluation, the revised version also incorporates an expanded cross-dataset validation against TermPicks, AutoTerm, CALFIN, and Black & Joughin (2023) using Average Minimum Distance (AMD)

metrics. Temporal interpolation between adjacent records is applied to address the scarcity of exact same-day overlaps, allowing near-coincident comparisons across the entire dataset. This approach provides a more representative and quantitative measure of overall dataset reliability.

26. Line 175 —"Its fine spatial detail and temporal coverage make it a valuable benchmark for evaluating glacier front delineations." That might be correct, but it is only available from 2014 onwards so that it is not useful for longer time series such as your dataset.

**Response:** We agree that PlanetScope imagery is only available from 2014 onward. Similar to the previous point, in the revised manuscript, its role is clarified as a quantitative assessment to confirm delineation precision. And the primary long-term validation is based on cross-comparison with multi-temporal products (TermPicks, AutoTerm, CALFIN, and Black & Joughin, 2023) using AMD metrics.

- 27. Line 177 Glacier names are incomplete and spelled wrong. *Response:* We thank the reviewer for pointing this out. All glacier names will be checked and corrected according to their official spellings based on NunaGIS and Biørk et al. (2015).
- 28. L183 "... capturing key features with sub-pixel accuracy..." if I understand this correctly you say that your delineations capture sub-pixel features. If so, I think that might just be coincidence as I don't think you'd be able to accurately delineate features within a pixel. Otherwise, please rephrase.

**Response:** The phrase "sub-pixel precision" referred to the delineations made relative to the imagery on which they were drawn, as verified against higher-resolution PlanetScope imagery. We agree that this wording was unclear and could be misinterpreted, so it has been revised to clarify that the delineations accurately capture the ice—ocean boundary at the resolution of the original source imagery.

**Text from the revised version (L183)**

In all six cases, the manually delineated fronts showed strong geometric agreement with PlanetScope imagery, accurately capturing the visible ice front—ocean boundary at the higher resolution imagery.

- 29. L188–189–I don't think the fact that this glacier is included in this dataset but not in others warrants the claim that this dataset has a greater spatial coverage. Especially since some of the other datasets do not contain land-terminating glaciers such as Kangilinguata Sermia. Consider rephrasing.
  - **Response:** We agree that the inclusion of this single glacier does not in itself justify a claim of broader spatial coverage. As noted above, this statement will be revised to clarify that the dataset complements existing products by incorporating additional glacier types—such as land- and lake-terminating glaciers—that are underrepresented by previous datasets.
- 30. **L207** Rephrase. I think you're trying to say that your dataset outperforms automated approaches but not all manually delineated datasets.
  - **Response:** Thank you for the suggestion. Once the dataset update is completed, we will perform a comprehensive cross-site comparison, and the corresponding statement will be adjusted accordingly.
- 31. Line 227: Make sure the citation is correct You are citing Goliber et al., (2022)

which is the publication, but you should be citing the dataset Goliber and Black, 2021)

**Response:** Thank you for your careful attention to detail. The citation has been corrected to reference the appropriate dataset — Goliber and Black (2021) — instead of the related publication (Goliber et al., 2022).

**32. 226–228** –I struggle to see how a close alignment to existing datasets is showing that your dataset is novel or more accurate. Maybe rephrase?

**Response:** The statement on higher accuracy referred specifically to comparisons with automated products, where our manually delineated fronts showed smaller deviations in complex terminus regions. The close agreement with other manually delineated datasets (e.g., TermPicks) instead demonstrates consistency and reliability, confirming that our dataset is accurate and compatible with established products rather than implying methodological novelty. This sentence has been revised.

**33.** L236 –It is unclear here if TermPicks of MEaSUREs is used as reference line. I assume that the AMD was calculated for bother and only the maximum is mentioned? Please be more specific.

**Response:** Yes, the AMD values shown in Figure 4 and related figures were calculated between this dataset and the most reliable available manual reference—TermPicks when available, and MEaSUREs otherwise. The reported values represent the mean differences across the entire calving front, which are slightly larger than the corresponding median values. This clarification has been added to the revised manuscript.

**34.** L279–280 –Replace "high positional consistency with" with "corresponds well with"

**Response:** Thank you for the suggestion. It has been replaced.

35. **L280-281**—"... depending on glacier geometry, image quality, and surface contrast." I would leave this out as you do not provide any measure on how any of these affect your accuracy.

**Response:** It has been removed.

36. Line 281: This is the first time semi-automated methods are mentioned. Unless it is introduced prior with a specific reference, I would leave it out.

**Response:** It has been removed.

**37.** L286–287 –From the figures in the manuscript, I struggle to see the increased spatial and temporal coverage compared to TermPicks. If anything, the dataset presented here seems to have less temporal coverage than existing datasets (e.g. Figure 7).

**Response:** The original spatial-coverage statistics were based on the total number of delineated glaciers, which exceed the glacier number of existing products. As noted above, we are refining the dataset's classification and expanding it to explicitly include more marine-, land-, lake-terminating, and floating-tongue glaciers. Once the updated dataset is completed, the total and marine- terminating glacier count will further increase. The corresponding figure will be revised and annotated to reflect the expanded coverage.

TermPicks and our datasets differ in scope and objectives, and they are complementary in terms of their spatial representation and temporal characteristics.

Regarding TermPicks, we agree that it provides excellent long-term temporal coverage for marine-terminating glaciers. Our dataset, however, has a broader scope, encompassing marine-, land-, and lake-terminating glaciers, and this coverage will be further expanded in the revised version. Our product is specifically designed to capture seasonal variability in calving-front positions. In the current version, most glaciers achieve at least two observations per year across the study period, and additional scenes are being incorporated to further enhance seasonal completeness. By contrast, TermPicks integrates results from multiple studies with varying temporal sampling. Its seasonal sampling reflects the availability of the underlying source datasets and therefore covers only a subset of glaciers and time periods, with variability among individual glaciers (Fig.2, Goliber et al. (2022)).

**Fig.2** Seasonal sampling examples from the TermPicks dataset for Graae Gletscher (116), Heinkel Gletscher (109), and Humboldt Gletsjer (85) (Goliber et al., 2022).

38. **L287–288** –Given that the dataset contains a multitude of glaciers that are landand lake-terminating and has no consistent temporal resolution, I would remove this sentence. Currently the dataset contains 227 marine-terminating glaciers, and the temporal resolution varies from annually to monthly.

**Response:** Thank you for this comment. As discussed above, the initial version of the dataset was indeed limited to glaciers within the Landsat-based velocity coverage and thus included a mix of marine-, land-, and lake-terminating types with variable temporal density. Following your suggestion, we are refining the dataset's scope and classification to explicitly distinguish glacier types and to standardize temporal resolution.

In the revised version, all glaciers will be clearly labelled by type (marine, land, lake, or floating-tongue) in the metadata, and the temporal sampling will be harmonized to a consistent seasonal interval where image availability allows. The sentence in question has been removed or rephrased accordingly to avoid overstatement of temporal uniformity.

**39.** L289–290 –It states here that there is "high-precision agreement with PlanetScope imagery" however earlier it is stated that this was only accessed visually. There is

no proof in the manuscript that would support the claim of "positional discrepancies are consistently below 10m". I suggest the authors rephrase or provide a detailed description of how this was determined.

**Response:** Thank you for pointing this out. In the revised manuscript, we have clarified the role of PlanetScope imagery and rephrased the corresponding statements. PlanetScope scenes are now used for multiple repeated measurements of calving fronts, where AMD values are computed directly to quantify the precision of the repeated manual delineations and, where possible, to allow quantitative comparisons with contemporaneous entries from other datasets. In addition, quantitative accuracy assessment is now addressed through an expanded AMD-based cross-dataset validation with TermPicks, AutoTerm, CALFIN, and Black & Joughin (2023), as detailed in the revised validation section.

40. **L301** -The name of the individual files "Calving\_front\_positions\_for\_ Greenland\_outlet\_glaciers\_2002\_2021 -New\_NeighborGID115" is unnecessarily long. I would recommend to either just use the Glacier ID or use the glacier name. In the current format, the key information (GlacierID) is almost hidden and I have to use half my screen in QGIS to see it.

**Response:** Thank you for this practical suggestion. The updated dataset no longer relies on glacier IDs. Instead, each glacier is identified by its official name according to map from NunaGIS and Bjørk et al. (2015), as the supplemented dataset also includes several non-marine-terminating glaciers not covered by existing ID.

**41.** L308 – The metadata should contain at least the official glacier name as specified in TermPicks or Bjork et al., (2015).

**Response:** The official glacier names will be added to the metadata according to map from NunaGIS and Bjørk et al. (2015).

**42.** L313 –The "season" field can be left out as it is quite clear from the "Date" field what season the delineation was taken in.

*Response:* It will be removed from the updated dataset.

43. L318 – Check the glacier names for spelling and consistency.

**Response:** They will be checked and corrected according to map from NunaGIS and Bjørk et al. (2015).

**44.** L**344** –Write these values as values±STD km.

**Response:** They will be expressed as mean  $\pm$  SD km.

- **45.** L350–351 –If you only model one glacier it's not really the dataset that you are using but a subset. I would leave this sentence out as it doesn't really add anything. *Response:* It has been removed.
- **46.** L376 –It should be mentioned here that the validation against PlanetScope imagery was visual only, and that the AMD was only determined for TermPicks and/or (?) MEaSUREs. Currently it reads like the validation of the dataset was extensive, which unfortunately, it was not.

**Response:** In the revised manuscript, this section has been rewritten to clarifying their respective purposes and scope. As noted in earlier responses, we are now conducting repeated measurements to quantify the precision of the manual delineations based om PlanetScope imagery. And an expanded cross-dataset validation that includes TermPicks, AutoTerm, CALFIN, and Black & Joughin

(2023) to provide a more systematic and transparent assessment of accuracy and representativeness.

We thank the reviewer again for the thorough and constructive feedback, and we hope that our responses and revisions fully address all concerns.

**Reference**

- Bjørk, A. A., Kruse, L. M., and Michaelsen, P. B.: Brief communication: Getting Greenland's glaciers right

   a new data set of all official Greenlandic glacier names, The Cryosphere, 9, 2215-2218,
  10.5194/tc-9-2215-2015, 2015.
- Dømgaard, M., Kjeldsen, K., How, P., and Bjørk, A.: Altimetry-based ice-marginal lake water level changes in Greenland, Communications Earth & Environment, 5, 365, 10.1038/s43247-024-01522-4, 2024.
- Goliber, S., Black, T., Catania, G., Lea, J. M., Olsen, H., Cheng, D., Bevan, S., Bjork, A., Bunce, C., Brough, S., Carr, J. R., Cowton, T., Gardner, A., Fahrner, D., Hill, E., Joughin, I., Korsgaard, N. J., Luckman, A., Moon, T., Murray, T., Sole, A., Wood, M., and Zhang, E. Z.: TermPicks: a century of Greenland glacier terminus data for use in scientific and machine learning applications, Cryosphere, 16, 3215-3233, 10.5194/tc-16-3215-2022, 2022.
- Kc, A., Enderlin, E. M., Fahrner, D., Moon, T., and Carroll, D.: Seasonality in terminus ablation rates for the glaciers in Greenland (Kalaallit Nunaat), The Cryosphere, 19, 3089-3106, 10.5194/tc-19-3089-2025, 2025.
- Rosenau, R., Scheinert, M., and Dietrich, R.: A processing system to monitor Greenland outlet glacier velocity variations at decadal and seasonal time scales utilizing the Landsat imagery, Remote Sensing of Environment, 169, 1-19, 10.1016/j.rse.2015.07.012, 2015.

---

## Author Comment (AC2)

**Response to review**

**Manuscript ID: ESSD-2025-304**

**Title:** Calving front positions for Greenland outlet glaciers (2002–2021): a spatially extensive seasonal record and benchmark dataset for algorithm validation

Dear Erik Loebel,

Thank you very much for your recognition and constructive comments. We fully accept all the suggestions and are incorporating them into both the manuscript and the dataset, and your feedback has been extremely helpful in guiding our ongoing revisions. Below is a summary of the main revisions followed by the point-to-point responses:

- 1. Clarified dataset coverage to include marine-, land-, lake-terminating, and floating-tongue glaciers, incorporated additional digitization to enhance spatial coverage, and added GlacierName and GlacierType fields.
- 2. Conduct a comprehensive cross-dataset Average Minimum Distance (AMD) assessment by identifying/interpolating at least one overlapping delineation per glacier and comparing it against TermPicks, AutoTerm, CALFIN, and Black & Joughin (2023).
- 3. Revised overly assertive wording, clarified the relevant limitations, and refined the dataset descriptions to improve clarity and consistency throughout the manuscript.

We hope that these revisions adequately address all your comments.

Best regards, Xi Lu and all co-authors

**General comments**

This paper presents a large dataset of manually delineated glacier calving fronts. These glacier fronts are valuable to the glaciology community, facilitating a better understanding of glacier dynamics and serving as constraints for ice dynamic modelling. They are also useful reference data for future machine learning-based delineation efforts. If I am not mistaken, this is currently the third largest manually delineated calving front dataset for Greenland (after the TermPicks repository and the data from Black and Joughin, 2023).

According to the manuscript, this product has (1) complete spatial coverage, (2) consistent and (3) high temporal resolution, as well as (4) homogeneous and (5) high accuracy delineation protocols. After reading the abstract and introduction, I anticipated a product that would meet all these criteria. However, upon reviewing the rest of the paper and examining the actual product, I was left quite disappointed, as only point 1 and, with some limitations, point 3 were actually met. I still believe this is a valuable

product that will benefit the glaciology community. As it stands, however, in its current form, the manuscript does not effectively describe the product, particularly its limitations. Significant revisions are required.

At various points in the paper (e.g. lines 19 and 78), it is emphasised that this data product has complete and consistent temporal sampling. This would be a major advantage, particularly for diversifying reference data for machine learning approaches, as most other manually delineated products have heterogeneous sampling, as they are usually by-products of focused glaciology studies. However, Figure 9 shows that the sampling is inconsistent. At L317, it is also stated that glaciers of higher scientific interest are captured at a higher temporal frequency, which seems to contradict the emphasis on temporal consistency somewhat. I am also not quite convinced by the definition of 'scientifically interesting'. Nioghalvfjerdsbræ only has 15 entries and Humboldt Glacier is not included at all. I don't see this as a major issue. I would just like these statements to be more representative of the actual product. Also, as this dataset is intended to serve as a benchmark, this should be mentioned as a limitation.

Response: We sincerely appreciate the reviewer's positive recognition of the value of our dataset to the glaciology community. Regarding the comments on the five key characteristics listed in the manuscript, we acknowledge that some statements in the original submission were overly positive and did not clearly articulate the limitations of our dataset. The corresponding explanations are provided below, and, in addition to supplementing the dataset where necessary, the revised manuscript presents the relevant limitations with greater accuracy and balance.

**(1) Complete spatial coverage:**

To support automated calving-front detection, our broader objective was to develop a dataset suitable for large-scale investigations of the seasonal behaviour of Greenland outlet glaciers, including joint analyses with seasonal velocity products derived from Landsat imagery over the past few decades. Accordingly, we used the spatial extent of glaciers covered by Landsat-based seasonal velocity fields as our selection criterion (Fig.1, Rosenau et al. (2015)). Glaciers outside this extent were assumed to exhibit limited frontal variability and were therefore not included. Within this defined spatial domain, our dataset effectively covers the glaciers (~290 compared to the 302 included in Rosenau et al. (2015), with some low-quality cases removed) for which seasonal velocity variations can be reliably detected.

Fig. 9. 302 locations (red circles) and names of all analysed outlet glaciers. The official spelling of the glacier names was chosen according to NunaGIS (2015). Unofficial glacier names are marked italic. The grey flow lines are determined by Gebler (2009). The hill-shading was performed with the elevation data from Ekholm (1996). The ice divide is highlighted with the red dotted line. The dashed blue lines indicate the regional boundaries.

**Fig.1** Selection basis of the research area, defined by the spatial extent of outlet glaciers covered by Landsat-derived seasonal velocity fields (Rosenau et al., 2015).

Considering the availability of existing datasets and the suggestions from the other reviewer, we have refined the dataset's definition of coverage to explicitly include marine-terminating, land-terminating, lake-terminating and floating-tongue glaciers, rather than limiting it to those defined by long-term seasonal velocity coverage. We are currently extending the dataset to incorporate additional glaciers of each type, which will be clearly labelled in the metadata to enhance accessibility and usability for the wider cryosphere research community.

**(2) Consistency and (3) High temporal resolution:**

The dataset was designed to achieve seasonal resolution (at least two scenes per year). For glaciers where GEEDiT imagery did not achieve this, additional images were manually acquired to maintain seasonal completeness. Compared with other published products, this dataset provides more complete seasonal coverage, especially during winter periods, by supplementing with Sentinel-1, ERS-1/2 and Envisat SAR imagery. The term "consistent" here was intended to describe this seasonal-scale fulfilment rather than perfectly uniform temporal sampling. Nevertheless, we agree that the temporal sampling still varies among glaciers due to image availability, and we have clarified this limitation in the revised manuscript.

**(4) Consistent and (5) High-accuracy delineation protocols:**

These terms were used to summaries the visual inspection and manual delineation procedures, which provide more reliable results than products with contributions from different authors and automated methods, particularly for glaciers with complex fronts. In the initial submission, these characteristics were supported by the six comparison results presented in the manuscript. In the revised version, we will provide additional methodological details and include expanded comparison results to further substantiate these two characteristics.

In addition to supplementing the dataset, we have softened the use of the terms *complete* and *consistent*, replacing them with *extensive spatial coverage* and *seasonally targeted sampling* to more accurately describe the dataset's characteristics. The revised description now defines the product as a high-precision, long-term seasonal record of calving front positions with broad coverage across diverse glacier types. The scope and limitations associated with each characteristic have been explicitly clarified throughout the manuscript.

Data quality and validation. This is the most critical point, in my opinion. The authors report an accuracy range of 40 to 100 metres for this data product. This value is derived by comparing the calving fronts in 15 satellite images across five glaciers to delineations from other data products. This is supplemented by visual comparisons against Planet imagery for another six calving fronts, as well as visual comparisons of five calving front change time series with other products.

The data quality of this product varies significantly from glacier to glacier (and also from delineation to delineation), but this is not picked up by the validation in the manuscript. It even feels somewhat disingenuous to show how well the glacier front's teeth-like structure is delineated in Fig. 4 when the quality of most other fronts is significantly worse. For example, many of the calving fronts of Nioghalvfjerdsbræ consist of fewer than 50 vertices for the entire 40 km-long glacier front, with some vertices being more than 5 km apart. These calving fronts clearly exceed 100 m in accuracy and are likely not suited for validating ML models. Similar issues are present for other glaciers. Clearly, validating using just six representative glaciers (with a total of 15 calving fronts assessed computationally) is insufficient to capture the

characteristics of the data product.

My suggestion is the following. Use the data from the larger, existing calving front repositories (I would recommend at least the four larger than this one, TermPick, AutoTerm, CALFIN and Black and Joughin, 2023). Identify pairs of calving fronts where there is an entry in both this and other data sets at the same date. Calculate the average minimum distance error between these two same-day entries. The results could then be shown as a data-product-to-data-product difference overall (e.g. as mean an median), and also per glacier (perhaps in a large table or histogram in the supplementary material). In conjunction with the validation results of the other data products, such an analysis would provide a much more complete picture of data quality.

This is only a recommendation, but I think this paper would benefit from a brief discussion of how it compares with other products. This could particularly cover other manual delineated repositories, such as TermPicks; other benchmark datasets, such as the one from Gourmelon et al. (10.5194/essd-14-4287-2022); and automation products, such as AutoTerm.

**Response:** Thanks for this important and constructive suggestion. In the initial submission, same-day delineations were difficult to identify for specific glaciers. Therefore, we selected one well-studied glacier from each basin to enable visual and quantitative comparisons. However, as both reviewers correctly noted, this approach was not sufficiently representative of the full dataset. For example, in Figure 4 the central section of the glacier front shows close agreement among all datasets and aligns well with the underlying imagery, while the lateral margins display larger differences, where the manual delineation appears to capture the visible shape more consistently than the automated products. Nevertheless, this single example does not capture the overall variability in delineation quality across Greenland.

Following the reviewer's comments, we are currently implementing a systematic cross-dataset validation. Our dataset is compared against the four largest existing calving-front products—TermPicks, AutoTerm, CALFIN, and Black & Joughin (2023)—using at least one same-day pair for each glacier included across the datasets. Where exact temporal matches are not available, the external products are interpolated to the closest available date to enable a consistent comparison. For overlapping calving fronts, we calculate the Average Minimum Distance (AMD) between corresponding front traces and summarize the results using the mean, median, and per-glacier distributions illustrated through histograms. Furthermore, in order to demonstrate the high precision of the data, we will also conduct repeat digitation on the same calving front and compare the results.

To further characterize delineation quality, we have also introduced a new vertex density (vertices per km) quality flag, enabling users to filter out sparsely sampled traces that may be unsuitable for machine-learning training or validation. While sparse

vertex distributions often reflect genuinely smooth and regular calving fronts rather than coarse delineation, this metric provides a transparent way to assess geometric detail.

In summary, the revised manuscript will incorporate a comprehensive cross-validation with all major existing datasets to provide a clearer and more representative assessment of data quality. Upon completion of the expanded comparison, the full distribution of inter-dataset differences will be presented using histograms across all glaciers. In addition, Section 2.3 will be updated to present a comprehensive comparison of our product against both manual and automated delineation datasets.

Glacier names and type (marine-terminating, land-terminating) should be included in the metadata. The writing in many places is below standard and requires revision. There are missing references (most notably in Table 1), places with citations where none is expected (e.g. L106, L117), mistakes (e.g. L140, Table 2) and inconsistencies (e.g. use of outlet and marine-terminating glacier). Please refer to the list of specific comments below.

Response: We appreciate the reviewer's careful reading and detailed corrections. The metadata will be expanded to include the fields GlacierName, defined with reference to the map from NunaGIS, and GlacierType, classified as marine-terminating, land-terminating, lake-terminating, or floating tongue. The manuscript has also been thoroughly revised to correct misplaced citations, add missing references (particularly in Table 1), and ensure consistency in terminology (e.g., "outlet glacier" vs. "marine-terminating glacier"). Furthermore, overall readability and language quality will be improved throughout. Detailed point-by-point responses to these specific issues are provided below.

**Response to Specific Comments**

1. L23 / L87 –Please provide an exact number.

*Response*: The exact numbers will be updated and reported in the revised manuscript after supplementing with additional glaciers (see response to major comment above).

2. L29 / L281 / L286 —I don't think the analysis in this manuscript fully justifies this statement. / This statement is not justified based on the analysis carried out here. / I don't see any advantage in terms of spatial or temporal coverage. TermPicks has similar spatial coverage and much longer temporal coverage.

*Response*: We agree that basing the analysis on just six representative glaciers in the initial submission was insufficient to fully justify the reported accuracy range. This limitation primarily arose from the difficulty of identifying overlapping same-day observations for the same glacier across datasets. The originally reported accuracy range of 40–100 m was therefore derived from a

small number of available sites and will be updated—and made more robust—using our expanded dataset and comprehensive cross-validation. Nevertheless, the initial results already indicate that our manual delineations are generally more reliable than automated products, particularly along complex glacier margins.

Our dataset and TermPicks differ in scope and objectives, but they are complementary in terms of their spatial representation and temporal characteristics. Regarding TermPicks, we agree that it provides excellent long-term temporal coverage for marine-terminating glaciers. Our dataset, however, has a broader scope, encompassing marine-, land-, and lake-terminating glaciers, and this coverage will be further expanded in the revised version. In addition, our product is specifically designed to capture seasonal variability in calving-front positions. In the current version, most glaciers achieve at least two observations per year across the study period, and additional scenes are being incorporated to further enhance seasonal completeness. By contrast, TermPicks integrates results from multiple studies with varying temporal sampling. Its seasonal sampling reflects the availability of the underlying source datasets and therefore covers only a subset of glaciers and time periods, with variability among individual glaciers (Fig.2, (Goliber et al., 2022)).

**Fig.2** Seasonal sampling examples from the TermPicks dataset for Graae Gletscher (116), Heinkel Gletscher (109), and Humboldt Gletsjer (85) (Goliber et al., 2022).

 L37 –Consider referring to the more recent IMBIE assessment (10.5194/essd-15-1597-2023) *Response*: The reference has been updated to include the more recent IMBIE (2023) assessment (doi:10.5194/essd-15-1597-2023).

4. L43 / L89 – Why specifically retreat?

Response: Our original intention was to refer to the overall retreating trend observed and projected for most Greenland outlet glaciers in previous studies. However, we agree that using "retreat" alone may imply a preconceived bias and is not fully objective. We have therefore rephrased it as "calving front migration" in the revised manuscript.

5. L65 –Black and Joughin (2023) did not use deep learning in this publication.

*Response:* We thank the reviewer for the correction. The reference to Black and Joughin (2023) has been removed from this sentence in the revised manuscript.

6. L70 – When speaking of 'current algorithms', why use a reference from 2011? Also, this statement is not entirely true.

*Response:* Thank you for pointing this out. We have updated the sentence to cite more recent studies and have revised the wording to provide a more accurate and balanced description of the capabilities and limitations of current delineation algorithms.

**Text from the revised version (L70)**

Recent studies show that automated calving-front algorithms would benefit from training data that better span diverse glacier geometries, have denser temporal sampling, include more complex surface conditions, and incorporate data from multiple sensors, thereby improving their applicability across regions (Cheng et al., 2021; Herrmann et al., 2023; Loebel et al., 2024).

7. L71 –This statement is misleading. I guess it only refers to the CALFIN product. The AutoTerm product has 278239 entries.

*Response:* We thank the reviewer for this clarification. In the initial submission, both CALFIN and AutoTerm (Zhang et al., 2023) were cited together because the latter also discussed similar limitations while proposing improvements. We have corrected the statement accordingly and now distinguish clearly between CALFIN and AutoTerm. The revised text now reflects the scale of existing products.

**Text from the revised version (L71)**

Despite being trained on more than 1,500 labelled fronts, advanced methods have extracted only ~22,000 Greenland calving-front positions (Cheng et al., 2021), a small fraction of the >400,000 available scenes. By incorporating the TermPicks dataset, Zhang et al. (2023) increased this to 278,239, demonstrating

that larger and higher-quality training data significantly improve model generalization, while indicating that intensive and more extensively sampled training sets are still required.

8. L61–84 –I feel like this paragraph needs to be completely restructured and given a common thread. The citations are all over the place. For example, why is Goliber et al. (2022, TermPicks) only mentioned in relation to the final statement and not for the product itself?

Response: We thank the reviewer for this valuable suggestion. The purpose of the original paragraph was to emphasise the need for reliable manually delineated datasets by outlining the limitations of existing automated approaches. As a result, most of the discussion focused on automated methods, and TermPicks, as a benchmark manual compilation, was only briefly mentioned at the end to highlight the importance of manual delineation. We agree that this structure weakened the narrative coherence and underrepresented the relevance of TermPicks.

In the revised manuscript, this paragraph will be substantially reorganized to improve its logical flow. TermPicks will be introduced earlier as a representative manual delineation dataset, discussed alongside automated products to provide direct comparison and context. This restructuring establishes a clearer link between manual and automated approaches and more effectively motivates the need for the new delineated product.

9. L81: We have products like this.

*Response*: We have reviewed and revised this sentence to more clearly highlight the distinctive features of this dataset, including its broad coverage across multiple glacier types and its seasonal temporal resolution.

Text from the revised version (L81)

Although several important products are now available, large-scale datasets that span diverse glacier types and offer broad seasonal sampling are still limited. This constrains the diversity of available training and verification data, and therefore reduces the opportunity to fully assess and further improve the generalization of emerging automated front-detection methods.

10. Table 1 –There are products missing: AutoTerm, ESA-CCI, https://doi.org/10.5194/essd-14-4287-2022, https://doi.org/10.5194/tc-17-1-2023, https://doi.org/10.5194/tc-18-3315-2024, https://doi.org/10.18739/A2W93G, https://doi.org/10.22008/FK2/UNZUJF, https://doi.org/10.5194/tc-12-3813-2018. Consider including a count of calving fronts. Isn't GEEDiT and ArcGIS manual as well? *Response:* We thank the reviewer for these suggestions. In the revised version, we will include the missing datasets (AutoTerm, ESA-CCI, Gourmelen et al., 2022, Choi et al., 2023, Ehrenfeucht et al., 2024, Brough et al., 2018, Miller et al., 2023, and Shen et al., 2018) in Table 1, and add an additional column showing the total number of calving fronts reported in each product.

Regarding the question about GEEDiT and ArcGIS, we clarify that both are used for manual delineation, but in slightly different ways. Manual refers to calving fronts digitized entirely by the user after independently downloading and preprocessing satellite imagery. GEEDiT, in contrast, streamlines this process by providing an interface for direct delineation without requiring prior image download, making it a more efficient semi-manual tool. We therefore distinguish between manual and GEEDiT approaches for clarity in the table.

11. L106 / L117 – Places with citations where none is expected (e.g. L106, L117), check these citations.

*Response:* We thank the reviewer for pointing this out. The misplaced citations at L106 and L117 have been removed in the revised manuscript.

12. Table 2 –Landsat 5 has an image resolution of 30 metres and no panchromatic band.

*Response*: Thank you for the correction. The Landsat 5 entry has been updated to reflect its 30 m resolution and lack of a panchromatic band. This was a table-preparation oversight and has now been corrected.

13. L140 –The NDWI is usually calculated using green and near-infrared. Why was red used instead?

Response: Thank you for pointing this out. We used the MOD09GQ product, which provides only the red and near-infrared bands at 250 m resolution. MODIS products that include the green band have much coarser spatial (~500–1000 m) or temporal (8-day) resolution and were therefore not suitable for our application.

14. L147 – Why grounded ice? What about floating glacier tongues?

Response: We thank the reviewer for this comment. In the initial submission, we referred to "grounded ice" because almost all glaciers in the dataset terminate in grounded fronts directly interacting with the ocean, with only two cases involving floating ice tongues. To ensure accuracy and inclusiveness, this has been revised to ice front, which also encompasses floating glacier tongues.

15. L149 –I guess only for Landsat-7 and 8?

*Response:* The text has been clarified to specify that the true-color composites generated within GEEDiT were used to provide clear visual contrast for manual delineation for optical imagery.

**Text from the revised version (L219-L221)**

For optical imagery, true-color composites generated in GEEDiT were used to enhance visual contrast for manual delineation.

16. L155 – Please refer to which glacier ID has been used.

*Response:* We appreciate the reviewer's suggestion. Following the comments from another reviewer, the updated dataset no longer relies on glacier IDs. Instead, each glacier is identified by its official name according to NunaGIS, as the update dataset will includes some non–marine-terminating glaciers not covered by existing ID.

17. L161 –I suspect this refers to the smallest possible error.

*Response*: We thank the reviewer for the clarification. The sentence has been revised to explicitly indicate that this refers to the resolution-limited lower bound, representing the smallest possible error.

18. L188 – From the satellite image in Figure 1, it looks like a land-terminating glacier. If that's true, that's probably why it hasn't been included in most other products. If that's the case, remove the subsequent statement.

*Response*: We thank the reviewer for this comment. As noted above, we have clarified the scope of the dataset, which includes multiple types of glaciers and is currently being expanded. Accordingly, Kangilinguata has been classified as a land-terminating glacier, and the corresponding description in Section 2.3 will be updated to reflect this adjustment.

19. L234 –Isstrøm is missing the "ø". Also in the figures. Please check.

*Response:* We thank the reviewer for pointing this out. All glacier names have been checked and corrected according to their official spellings based on map from NunaGIS and Bjørk et al. (2015), including the proper use of "ø" in Isstrøm and other figures.

20. L304 – This refers to the .shp files, whereas the product itself is a Geopackage (.gpkg).

*Response:* We appreciate the reviewer's correction. The product was originally submitted in shapefile (.shp) format and later converted to GeoPackage (.gpkg)

following the editor's comments. This text was overlooked during the format update and has now been corrected.

21. Figure 10: How is this calculated? What was done with Humboldt Glacier, which is not included in the product?

Response: The values shown in Figure 10 were derived by averaging the 20-year calving-front change along the central flowline for all glaciers within each major drainage basin. Prior to submission, we removed several preliminary or lower-confidence delineations to ensure consistency and reliability across the full dataset. Humboldt Glacier was among the exclusions because its exceptionally wide front and highly fragmented northern margin generated irregular lateral endpoints that could not be standardized to a dataset-wide consistency level in the initial release. In the revised version, we will reprocess and include Humboldt Glacier to improve the spatial completeness of the dataset.

22. L350–359 –This does not really fit in this section, which should focus on the product and usage notes.

*Response*: We thank the reviewer for the comment. The paragraph describing the ISSM modeling application has been removed to keep this section focused on the data product and usage notes.

We hope the changes we have made address all points raised and bring the submission in line with ESSD's technical and editorial standards. Please let us know if further clarification or modification is needed.

**References**

- Bjørk, A. A., Kruse, L. M., and Michaelsen, P. B.: Brief communication: Getting Greenland's glaciers right

   a new data set of all official Greenlandic glacier names, The Cryosphere, 9, 2215-2218,
  10.5194/tc-9-2215-2015, 2015.
- Goliber, S., Black, T., Catania, G., Lea, J. M., Olsen, H., Cheng, D., Bevan, S., Bjork, A., Bunce, C., Brough, S., Carr, J. R., Cowton, T., Gardner, A., Fahrner, D., Hill, E., Joughin, I., Korsgaard, N. J., Luckman, A., Moon, T., Murray, T., Sole, A., Wood, M., and Zhang, E. Z.: TermPicks: a century of Greenland glacier terminus data for use in scientific and machine learning applications, Cryosphere, 16, 3215-3233, 10.5194/tc-16-3215-2022, 2022.
- Rosenau, R., Scheinert, M., and Dietrich, R.: A processing system to monitor Greenland outlet glacier velocity variations at decadal and seasonal time scales utilizing the Landsat imagery, Remote Sensing of Environment, 169, 1-19, 10.1016/j.rse.2015.07.012, 2015.